# CLiMB: A Continual Learning Benchmark for Vision-and-Language Tasks

**Tejas Srinivasan**[1]     **Ting-Yun Chang**[1]     **Leticia Pinto Alva**[1]
**Georgios Chochlakis**[1]     **Mohammad Rostami**[1,2]     **Jesse Thomason**[1]
[1]University of Southern California     [2]USC Information Sciences Institute
{tejas.srinivasan,tingyun,pintoalv,chochlak,rostamim,jessetho}@usc.edu

## Abstract

Current state-of-the-art vision-and-language models are evaluated on tasks either individually or in a multi-task setting, overlooking the challenges of continually learning (CL) tasks as they arrive. Existing CL benchmarks have facilitated research on task adaptation and mitigating "catastrophic forgetting", but are limited to vision-only and language-only tasks. We present CLiMB, a benchmark to study the challenge of learning multimodal tasks in a CL setting, and to systematically evaluate how upstream continual learning can rapidly generalize to new multimodal and unimodal tasks. CLiMB includes implementations of several CL algorithms and a modified Vision-Language Transformer (ViLT) model that can be deployed on both multimodal and unimodal tasks. We find that common CL methods can help mitigate forgetting during multimodal task learning, but do not enable cross-task knowledge transfer. We envision that CLiMB will facilitate research on a new class of CL algorithms for this challenging multimodal setting.

## 1  Introduction

Large-scale pre-trained models, including crossmodal vision-and-language models, are generally fine-tuned on each downstream task individually, requiring fine-tuning and storing new models for each task. By contrast, multi-task learning requires fixing a set of tasks, but such training is incapable of dynamically learning new tasks. Although continual learning (CL) algorithms have explored cross-task knowledge transfer, existing methods primarily consider unimodal tasks in artificial settings [Jin et al., 2021, Lin et al., 2021]. Multimodal pre-training can encode useful and transferable features for diverse tasks, but learning from a *sequence* of different multimodal tasks and the subsequent forgetting effects [Kirkpatrick et al., 2017] have not yet been studied.

Additionally, it is assumed that these deployed models will encounter tasks containing all modalities seen during training time. This assumption means learning separate models for language-only, vision-only, and vision-language tasks, as opposed to a single "generalist" model that can handle all modalities or subsets of them [Reed et al., 2022]. Yet, existing work suggests that knowledge grounded in multiple modalities can benefit unimodal tasks [Desai and Johnson, 2021, Jin et al., 2022]. Currently, the research community lacks a suitable benchmark to systematically study how models can continually learn vision-and-language tasks while being transferable to unimodal tasks.

In this paper, we introduce the **Continual Learning in Multimodality Benchmark (CLiMB)**,[1] to facilitate the study of CL in vision-and-language tasks with deployment to multi- and unimodal tasks. We formulate a learning problem wherein a model is first trained on sequentially arriving vision-and-language tasks, referred to as **upstream continual learning**, and then **transferred downstream to low-shot** multimodal and unimodal tasks. CLiMB initially includes four vision-and-language

---

[1]The code for our benchmark is available at `https://github.com/GLAMOR-USC/CLiMB`

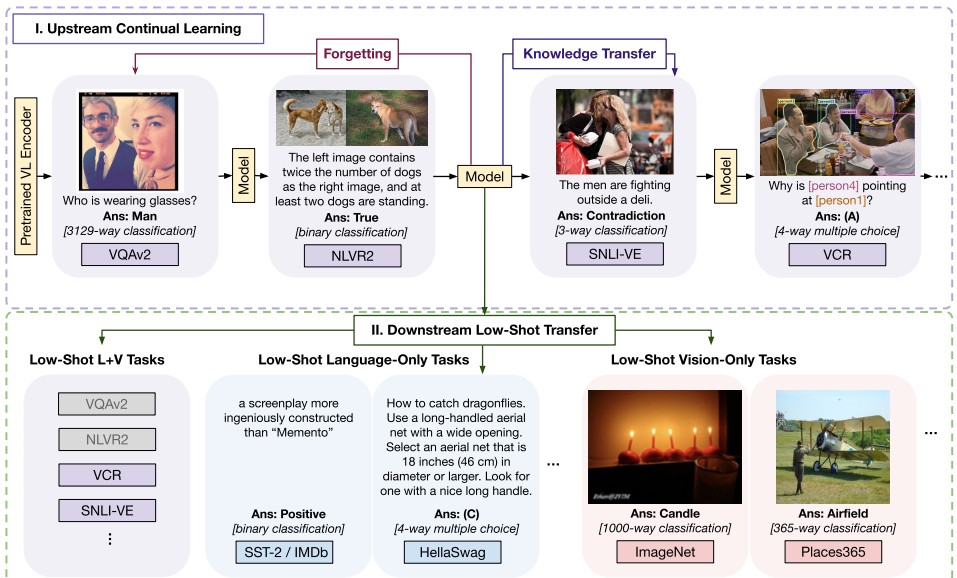

Figure 1: CLiMB evaluates candidate CL models and learning algorithms in two phases. For Phase I, Upstream Continual Learning, a pre-trained multimodal model is trained on a sequence of vision-and-language tasks, and evaluated after each task on its degree of Forgetting of past tasks and Knowledge Transfer to the next task. For Phase II, after each multimodal task the model is evaluated for its Downstream Low-Shot Transfer capability on both multimodal and unimodal tasks.

tasks, five language tasks, and four vision tasks, and is extensible to new tasks, models, and learning algorithms. Experiments using CLiMB find that existing CL algorithms can mitigate forgetting, but not transfer knowledge across tasks, revealing a need for new research into continual learning strategies for vision-language tasks. Further, current CL algorithms and multimodal models are not well suited for low-shot adaptation to multimodal or unimodal tasks. We hope CLiMB will provide the basis for developing models and learning algorithms for multimodal continual learning.

## 2   Background and Related Work

Continual, or lifelong, learning is an essential ability to develop autonomous agents that can learn in a cumulative way [Chen and Liu, 2018]. In CL, a model is trained on sequentially arriving tasks and evaluated both on its ability to learn future tasks as well as retain performance on past learned tasks [Kirkpatrick et al., 2017]. A necessity for developing CL algorithms is benchmarks that collate suitable sequential tasks. There are two primary approaches to create such CL benchmarks.

The first approach is to split existing tasks into non-overlapping sub-tasks that are sequentially presented for continual learning. For example, one can divide tasks along input categories [Greco et al., 2019] or output classes [Vinyals et al., 2016, Kirkpatrick et al., 2017] into disjoint sets. Mimicking real world distribution shift, timestamps can group data instances according to the order of their creation [Lin et al., 2021].

CLiMB goes beyond such artificial, single-task-based CL and instead aggregates several diverse tasks. Similarly, unimodal benchmarks such as Visual Domain Decathlon [Rebuffi et al., 2017] and Natural Language Decathlon [McCann et al., 2018] aggregate 10 image classification and 10 language tasks, respectively. The CLIF-26 benchmark [Jin et al., 2021] is built for CL on the GLUE [Wang et al., 2019] language tasks. CLiMB goes beyond these unimodal benchmarks by evaluating on sequences of multimodal, vision-and-language tasks *and* testing downstream transfer to unimodal tasks.

| Task | Vision Input(s) | Language Input(s) | Decision | Score Metric |
|------|-----------------|-------------------|----------|--------------|
| VQAv2 | Image | Question | 1 of 3129 | VQAScore[2] |
| NLVR2 | 2 images | Caption | True/False | Accuracy |
| SNLI-VE | Image | Hypothesis | Ent/Neu/Con | Accuracy |
| VCR | Image w/ bboxes | Question + 4 Answers | 1 of 4 | Accuracy |
| IMDb | | Sentence | Pos/Neg | Accuracy |
| SST-2 | | Sentence | Pos/Neg | Accuracy |
| HellaSwag | | Sentence Prefix + 4 Endings | 1 of 4 | Accuracy |
| CommonsenseQA | | Question + 5 Answers | 1 of 5 | Accuracy |
| PIQA | | Question + 2 Answers | 1 of 2 | Accuracy |
| ImageNet-1000 | Image | | 1 of 1000 | Top-1 Accuracy |
| iNaturalist2019 | Image | | 1 of 1010 | Top-1 Accuracy |
| Places365 | Image | | 1 of 365 | Top-1 Accuracy |
| COCO-object | Image | | $n$ of 80 | Micro-F1 |

Table 1: The initial tasks in CLiMB include various forms of vision and language inputs, and each task is framed as a classification problem. Multimodal vision-and-language tasks serve as upstream training for both multimodal and unimodal downstream, low-shot tasks.

## 3 CLiMB: The Continual Learning in Multimodality Benchmark

CLiMB tests the ability of models and learning algorithms to adapt to a sequentially arriving stream of vision-language tasks, as well as rapidly transfer to new multimodal and unimodal tasks (Table 1).

### 3.1 CLiMB Learning and Evaluation

Learning and evaluation in CLiMB proceeds in two phases: **upstream continual learning** and **downstream low-shot transfer** (Figure 1). Table 2 summarizes our CL evaluation metrics. We denote a task with modality $M \in \{V, L, VL\}$ as $\mathcal{T}_M^i$ and the number of such tasks as $K_M$.

**Upstream Continual Learning of Multimodal Tasks**    A candidate model $\mathcal{M}$ encounters a sequence of $K_{VL}$ vision-language tasks, $\mathcal{T}_{VL}^{1...K_{VL}}$. $\mathcal{M}$ can be initialized with a pre-trained encoder. We allow parameter additions to the base model on a per-task basis. In this work we add task-specific classification layers for each new task on top of the base encoder model. The model $\mathcal{M}$ is sequentially trained on the training split of each task $\mathcal{T}_{VL}^i$ with candidate CL algorithm $\mathcal{A}$. For task $\mathcal{T}_{VL}^i$, the model is not presented with any inputs from $\mathcal{T}_{VL}^{1...i-1}$. However, the CL algorithm $\mathcal{A}$ may allocate memory to access previous training examples.

We evaluate two primary model properties in the upstream phase: **upstream knowledge transfer** from past learned tasks to new tasks, and withstanding **forgetting** of previously-seen tasks. The upstream knowledge transfer $\mathbb{T}_{UK}(i)$ on task $\mathcal{T}_{VL}^i$ is the relative gain in score from learning the previous tasks $\mathcal{T}_{VL}^{1...i-1}$. Forgetting $\mathbb{T}_F(j \leftarrow i)$ of previously-seen task $\mathcal{T}_{VL}^j$ is the relative performance degradation in that task after learning subsequent tasks $\mathcal{T}_{VL}^{j+1...i}$ (Table 2).

**Downstream Transfer to Low-Shot Tasks**    We evaluate the low-shot adaptation ability of the model $\mathcal{M}$ after learning each upstream vision-language task. After training on the $i^{th}$ upstream task $\mathcal{T}_{VL}^i$, we evaluate low-shot transfer to remaining multimodal tasks $\mathcal{T}_{VL}^{i+1...K_{VL}}$, as well as unimodal tasks $\mathcal{T}_V^{1...K_V}$ and $\mathcal{T}_L^{1...K_L}$. Specifically, for every task in each modality, a low-shot instance of task $\mathcal{T}_M^i$, denoted as $\mathcal{T}_M^{LS(i)}$, is created. The **low-shot transfer** ability to this task is evaluated by fine-tuning upstream encoder checkpoints on task $\mathcal{T}_M^{LS(i)}$. We compute the low-shot transfer $\mathbb{T}_{LS}^M(i)$ as the relative improvement of the CL encoder's performance on the low-shot task $\mathcal{T}_M^{LS(i)}$, denoted as $S_{\mathcal{A}}^{LS(i)}$, over the pre-trained encoder's performance on the same low-shot task, $S_{PT}^{LS(i)}$.

---

[2]https://visualqa.org/evaluation.html

| Evaluation Type | Description | Metric ($\times 100\%$) |
|---|---|---|
| Upstream Knowledge Transfer, $\mathbb{T}_{UK}(i)$ | Improvement of performance on task $\mathcal{T}_{VL}^i$ after training on tasks $\mathcal{T}_{VL}^{1\cdots i}$ using algorithm $\mathcal{A}$ ($S_{\mathcal{A}}^i$) compared to finetuning the pretrained model on $\mathcal{T}_{VL}^i$ directly ($S_{PT}^i$). | $\mathbb{T}_{UK}(i) = \frac{S_{\mathcal{A}}^i - S_{PT}^i}{S_{PT}^i - S_R^i}$ |
| Forgetting Transfer, $\mathbb{T}_F(j \leftarrow i)$ | Decrease of performance when a model trained on tasks $\mathcal{T}_{VL}^{1\cdots i}$ is evaluated on task $\mathcal{T}_{VL}^j$ ($j < i$). $S_{\mathcal{A}}^{j \leftarrow i}$ denotes model performance on $\mathcal{T}_{VL}^j$ after training up to $i$. | $\mathbb{T}_F(j \leftarrow i) = \frac{S_{\mathcal{A}}^j - S_{\mathcal{A}}^{j \leftarrow i}}{S_{\mathcal{A}}^j - S_R^j}$ |
| Low-Shot Transfer, $\mathbb{T}_{LS}^M(i)$ | Improvement of performance on low-shot task $\mathcal{T}_M^{LS(i)}$ using an encoder checkpoint trained by upstream algorithm $\mathcal{A}$ ($S_{\mathcal{A}}^{LS(i)}$) compared to learning the same low-shot task without any upstream learning ($S_{PT}^{LS(i)}$). | $\mathbb{T}_{LS}^M(i) = \frac{S_{\mathcal{A}}^{LS(i)} - S_{PT}^{LS(i)}}{S_{PT}^{LS(i)} - S_R^i}$ |

Table 2: Model and algorithm evaluation metrics in the upstream and downstream phases. For the $i^{th}$ task, we compute a model's $\delta^i = S^i - S_R^i$, the model's task score $S^i$ minus the score $S_R^i$ of random selection. Our evaluation protocol computes each metric as a relative change in the $\delta^i$ of a CL algorithm $\mathcal{A}$ over a baseline setting $\mathcal{B}$ to enable across-task comparisons. Each evaluation metric is calculated as $\frac{\delta_{\mathcal{A}} - \delta_{\mathcal{B}}}{\delta_{\mathcal{B}}} = \frac{S_{\mathcal{A}} - S_{\mathcal{B}}}{S_{\mathcal{B}} - S_R}$, and is presented as a percentage.

## 3.2 CLiMB Multimodal and Unimodal Tasks

CLiMB initially includes four multimodal upstream vision-language tasks, five language-only tasks, and four vision-only tasks (Table 1). We frame each as a classification task.

**Vision-Language Tasks**   CLiMB includes VQAv2 [Goyal et al., 2017], NLVR2 [Suhr et al., 2019], SNLI-VE [Xie et al., 2019] and VCR [Zellers et al., 2019a]. Solving these challenging tasks requires different kinds of knowledge in the multimodal model: question answering, visual and commonsense reasoning, entailment understanding.

**Language-Only Tasks**   CLiMB includes IMDb [Maas et al., 2011], SST-2 [Socher et al., 2013], HellaSwag [Zellers et al., 2019b], CommonsenseQA [Talmor et al., 2019], and PIQA [Bisk et al., 2020]. We hypothesize that visually-grounded knowledge from upstream tasks may benefit tasks such as IMDb and SST-2, which are sourced from movie reviews, as well as HellaSwag, sourced from video captions, and PIQA, sourced from physically-grounded instructions from images and videos. Further, commonsense knowledge and reasoning skills obtained from VCR and NLVR2 may benefit tasks like HellaSwag, CommonsenseQA, and PIQA.

**Vision-Only Tasks**   CLiMB includes ImageNet-1000 [Russakovsky et al., 2015], iNaturalist2019 [Van Horn et al., 2018], Places365 [Mahajan et al., 2018], and MS-COCO object detection (formulated as a multi-label classification task). Since VQAv2 images are sourced from MS-COCO [Lin et al., 2014], we hypothesize VQAv2 upstream learning may aid in the COCO object detection task.

## 4   Modeling and Experiments

Using CLiMB, we study the performance of several commonly used CL algorithms on multimodal tasks. We use fixed upstream task order (VQAv2 → NLVR2 → SNLI-VE → VCR).

### 4.1   Vision-Language Encoder: ViLT

We use a pre-trained Vision-Language Transformer (ViLT) [Kim et al., 2021] as a backbone encoder. Unlike other pre-trained vision-language models [Lu et al., 2019, Chen et al., 2020] that build upon region-level features extracted from Faster R-CNN [Ren et al., 2015], ViLT directly operates on image patches without using any convolutional layers. In ViLT, text tokens and image patches are concatenated into an input sequence and passed through a Transformer, which learns the vision-language alignment with self-attention across both modalities (Figure 2).

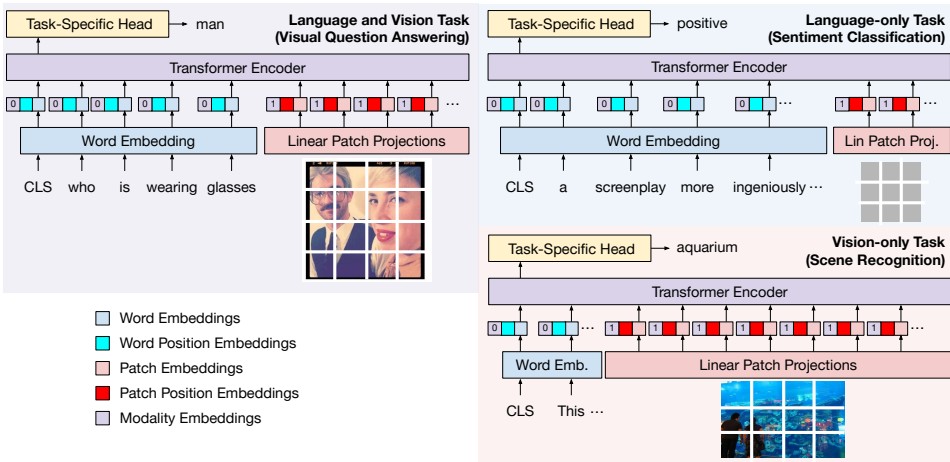

Figure 2: The ViLT model [Kim et al., 2021] operates on vision and language inputs (left). We adapt these inputs for language-only tasks by providing the average MS-COCO image as in-domain visual input, and vision-only tasks by providing vacuous language input "This is an image" (right).

## 4.2 Upstream Experiments: Algorithms and Task Ordering

CLiMB includes several CL algorithm implementations. **Sequential Fine-tuning (SeqFT)** fine-tunes the full encoder and task-specific layers for each task in order. This baseline algorithm is an extension of the single-task fine-tuning paradigm to the CL setting. We also experiment with a **Frozen Encoder** baseline that trains only the task-specific layers. Fine-tuning all parameters may cause forgetting since the encoder parameters are overwritten, while fine-tuning only the task-specific layer prevents knowledge transfer since the shared encoder parameters are fixed. In **Frozen Bottom-K**, we freeze the bottom $K$ layers of the encoder and fine-tune the rest, balancing these solutions (we set $K$=9).

CLiMB also includes two CL algorithms that fine-tune all parameters but are designed to mitigate forgetting. **Experience Replay (ER)** [Chaudhry et al., 2019] caches a small percentage of training examples in a memory buffer after training on each task, and periodically performs a "replay" training step using cached examples to refresh the model. **Elastic Weight Consolidation (EWC)** [Kirkpatrick et al., 2017] is a regularization method that adds an auxiliary L2 loss between weights in the current model and previous checkpoints to slow change in important encoder parameters.

Finally, CLiMB includes **Adapters** [Houlsby et al., 2019], which add a small number of task-specific parameters, called Adapter modules, within layers of the pre-trained encoder. During training, the encoder's original parameters are kept frozen and only the Adapter modules are trained. We use a new set of Adapter modules each time we train on a new task, which leaves the previous tasks' modules untouched and prevents forgetting, but also does not facilitate cross-task knowledge transfer.

## 4.3 Downstream Low-Shot Experiments

We consider low-shot multimodal and unimodal tasks. We first define low-shot settings for different task types, then explain how we apply the multimodal ViLT model to unimodal settings.

**Low-Shot Task Settings** We study "low-shot" training paradigms where only a fraction of full training data is available. For the multimodal classification tasks, NLVR2 and SNLI-VE, we use 2048 examples per class, whereas for the multiple choice VCR task, we use 5% of the training data. Among unimodal tasks, for vanilla classification tasks (IMDb, SST2, ImageNet-1000, iNaturalist, and Places365), we consider training with $N = \{16, 32\}$ examples per class. For multiple choice classification tasks (PIQA, CommonsenseQA, HellaSwag), we use $N = \{1024, 4096\}$ since these tasks are considerably more challenging. For the multi-label COCO object detection task, we consider $M = \{5\%, 10\%\}$ of the original training data.

| Alg $\mathcal{A}$ | Params Trained | Task 1 VQAv2 | Task 2 NLVR2 | Task 3 SNLI-VE | Task 4 VCR |
|---|---|---|---|---|---|
| Direct FT | 100% | [67.70] | [73.07] | [76.31] | [61.31] |
| SeqFT | 100% | 0.13% [67.79] | -1.80% [72.66] | -3.33% [74.89] | -5.09% [59.47] |
| Frozen Enc | 7.88% | -14.10% [58.15] | -40.78% [63.66] | -15.98% [69.45] | -53.47% [41.90] |
| Frozen B9 | 25.92% | -0.58% [67.30] | -0.58% [72.94] | -3.31% [74.90] | -15.49% [55.69] |
| ER | 100% | 0.26% [67.87] | 0.56% [73.20] | -2.89% [75.08] | -4.45% [59.70] |
| EWC | 100% | 0.20% [67.84] | -2.79% [72.39] | -4.52% [74.38] | -4.86% [59.55] |
| Adapters | 13.02% | **0.59% [68.10]** | **2.55% [73.66]** | **-0.56% [76.08]** | **-0.36% [61.18]** |

Table 3: Upstream Knowledge Transfer $\mathbb{T}_{UK}(i)$ relative to direct fine-tuning on each task, along with task score $[S_{\mathcal{A}}^i]$ (%), for different CL algorithms $\mathcal{A}$ applied to ViLT. No CL algorithms achieve notable positive Knowledge Transfer, while the majority in fact *hurt* learning of new tasks.

**Unimodal Tasks in ViLT**  To apply ViLT to vision-only tasks, we use the phrase "This is an image" as the language input paired with the input image from the vision task. For language-only tasks, however, we need to address several challenges to effectively apply ViLT.

First, we find that averaging all MS-COCO training images into a single, in-distribution image paired with text inputs produces better results with ViLT than not concatenating any image tokens at all to the Transformer input sequence.

Second, ViLT only allows a maximum of 40 language tokens in the input, which is enough for captions but insufficient for most language tasks. To deal with this challenge, we first downsample the vacuous image to reduce its token length from 144 to 16. Next, we extend the available language tokens by creating copies of pre-trained ViLT's language positional embeddings, $E \in \mathbb{R}^{40 \times d}$, and concatenating these copies to get extended positional embeddings, $\hat{E} \in \mathbb{R}^{L \times d}$, where $L$ is the maximum sequence length of each task and $d$ is the embedding dimension.

Finally, ViLT's language pre-training is on image captions that do not represent more general language use. We additionally experiment with a VAuLT [Chochlakis et al., 2022] model that extracts language representations from a pre-trained, frozen BERT [Devlin et al., 2019] that serve as input embeddings for ViLT. Please refer to the supplementary materials for more experiments and details.

# 5 Results

We present Knowledge Transfer and Forgetting capabilities of different CL algorithms, experiments with multiple upstream task orders, and Low-Shot Transfer to downstream tasks.

## 5.1 Upstream Learning Results

We find that common CL algorithms do not facilitate positive knowledge transfer in the vision-and-language setting of CLiMB, and in fact often hurt future task learning. Some are able to effectively mitigate forgetting, but none perform as well as directly fine-tuning on a candidate task. By examining the effects of task order, we conclude that the VCR task hurts further upstream task learning.

**Upstream Knowledge Transfer**  In Table 3, we compare the upstream knowledge transfer exhibited by the different algorithms described in Section 4.2. Freezing the entire encoder severely under-performs the direct fine-tuning baseline for each task. Among other methods, all perform similarly to directly fine-tuning on the first task, with approximately zero knowledge transfer. However, for all methods other than Adapters, more continual learning hurts the model's ability to learn new tasks, as evidenced by the increasingly negative upstream transfer for later tasks. This effect may be due to loss of pre-training knowledge which is useful for task adaptation. This property of models to learn new tasks poorly in a continual learning setting is also called intransigence [Chaudhry et al., 2018]. Adapters, which do not train shared encoder parameters, do not exhibit this negative knowledge transfer, and show comparable performance to full model fine-tuning despite having very few learnable parameters.

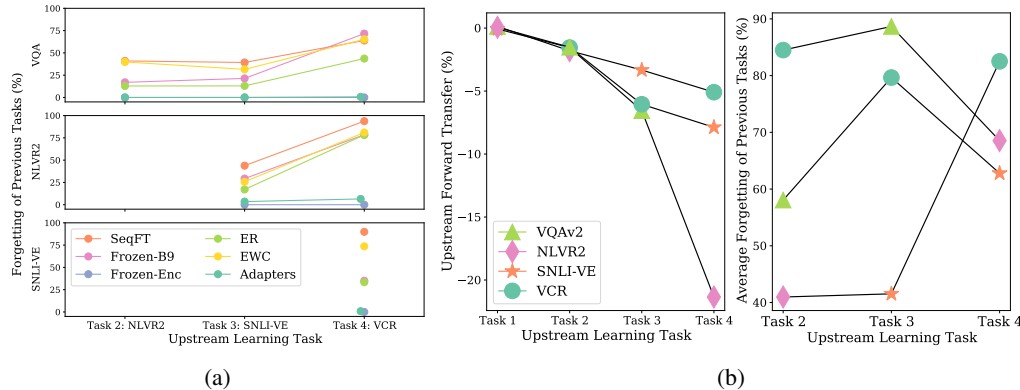

(a)                                          (b)

Figure 3: **(a)** Forgetting $\mathbb{T}_F(j \leftarrow i)$ (%) of the previous $i-1$ tasks for each algorithm. Each subplot denotes model performance on one of the previous tasks. While all algorithms that fine-tune shared parameters exhibit Forgetting, ER best preserves past task performance. **(b)** Effect of task order on upstream Knowledge Transfer (left) and Forgetting (right) for three different orders. Lines represent performance conditioned on a particular task order. After experiencing the VCR task, models exhibit lower Knowledge Transfer and higher Forgetting.

**Forgetting** Figure 3a shows how each algorithm affects forgetting of previous tasks. Sequential Fine-tuning forgets previous tasks to a large extent, Frozen Bottom-9 shows slight improvement, and freezing the encoder prevents forgetting entirely. Experience Replay does a better job at retaining task performance, while EWC shows only a slight improvement. Adapters enable a model to learn upstream tasks in the multimodal CL setting *while not forgetting tasks already learned*, adding only 3-4% parameters per task. Interestingly, forgetting is more severe after training models on the VCR task, demonstrating that the order of encountering tasks affects continual learning.

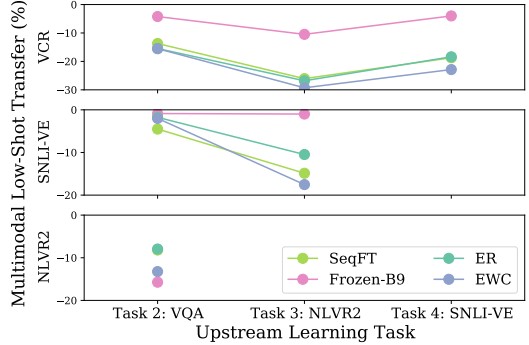

Figure 4: Low-shot transfer, $\mathbb{T}_{LS}^{VL}(j)$, for multimodal tasks $j = \{i+1, ..., K_{VL}\}$ after training on upstream task $\mathcal{T}_{VL}^i$. All CL algorithms exhibit negative Low-shot transfer on all multimodal tasks.

**Effect of Upstream Task Order** Figure 3b shows the upstream knowledge transfer and forgetting for ViLT using Sequential Fine-tuning on three different task orders. While the upstream transfer is similar for the first two tasks in each task ordering, training on VCR negatively affects both knowledge transfer to future tasks and forgetting of past tasks. This effect may be due to a shift in the visual domain of VCR: input images have colored boxes drawn on them to represent grounded objects in the question, following previous work [Zellers et al., 2021, Hessel et al., 2022].

## 5.2 Downstream Low-Shot Transfer Results

In downstream transfer, we fine-tune the entire model irrespective of the upstream CL algorithm. We find that upstream learning with current CL algorithms[3] does not help the ViLT encoder generalize to multimodal and unimodal tasks in low-shot settings.

**Vision-Language Tasks** Figure 4 presents the low-shot transfer $\mathbb{T}_{LS}^{VL}(j)$ for all future tasks $j > i$ after training on upstream task $\mathcal{T}_{VL}^i$ (x-axis). We observe that low-shot transfer is always negative, implying that upstream continual learning always hurts the model's ability to learn new tasks in low-shot settings. Since upstream learning hurts model adaptation on new multimodal tasks with full training data (Table 3), it is expected that this effect will also be reflected in the low-shot regime.

---

[3]We do not include Adapters and Frozen-Encoder as they do not modify pre-trained ViLT's parameters.

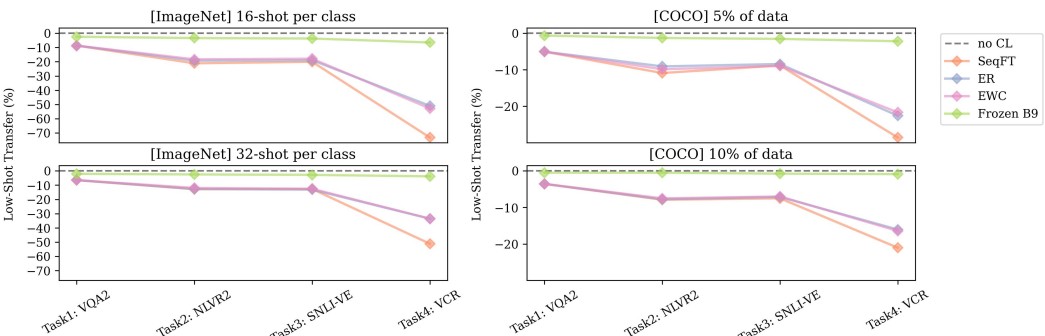

Figure 5: Low-Shot Transfer (%) comparison between different CL algorithms on downstream vision-only tasks (left: ImageNet; right: COCO). Findings on iNaturalist 2019 and Places365 are similar to ImageNet (see Supp). Generally, current CL algorithms hurt low-shot transfer compared to direct fine-tuning, with Frozen Bottom-9 being the least harmful.

**Vision-Only Tasks**   Figure 5 presents low-shot transfer on vision downstream tasks, using checkpoints from different upstream CL algorithms. Fine-tuning ViLT without CL performs well on vision-only tasks, achieving $65\%$ top-1 accuracy on ImageNet-1000 with only 16 shots per class (see Supp). This performance suggests that ViLT already contains rich visual representations, making it sample efficient when transferred to vision-only tasks.

Second, CL actually *hurts* the transferability to downstream vision tasks. Among CL algorithms, Sequential Fine-tuning is the most harmful one, while freezing the bottom 9 layers causes the least degradation, almost matching direct fine tuning. This finding is consistent with previous work suggesting that bottom layers in deep models learn more general and transferable representations than upper layers [Yosinski et al., 2014, Lee et al., 2019].

Notably, upstream VQA and SNLI-VE checkpoints have a less negative effect on downstream COCO performance compared to NLVR2 and VCR. Because images from NLVR2 and VCR are more dissimilar to MS-COCO than the image sources of VQA and SNLI-VE, we hypothesize that large data distribution shifts between upstream and downstream tasks hurts transfer.

**Language-Only Tasks**   In Figure 6, we compare the performance of two pre-trained encoders, ViLT and VAuLT, on low-shot language tasks, and the effects of upstream multimodal CL on low-shot transfer when applied to both encoders.

We observe that model adaptation to language tasks is challenging. The ViLT model frequently performs only marginally better than the random baseline, regardless of the upstream algorithm. Using VAuLT as the encoder achieves notably higher accuracy compared to ViLT on all tasks, indicating that strong language priors are key to low-shot language adaptation.

All upstream CL tasks improve VAuLT's transferability to SST-2 except for VCR. For both SST-2 and IMDb, there are significant drops after learning VCR in the upstream phase with ViLT and VAuLT, following vision-only results showing VCR is farther out of distribution than other upstream tasks.

However, we do not observe similar trends on the three multiple-choice tasks, where CL generally hurts. We believe that current multimodal tasks do not learn complex language reasoning skills, hurting model transferability to language-only reasoning tasks.

## 6   Conclusions

We propose the **Continual Learning in Multimodality Benchmark (CLiMB)** to study CL in multimodal tasks with deployment to multi- and unimodal tasks. Our experiments find that existing CL strategies do not generalize well to sequences of multimodal tasks or enable effective low-shot adaptation to downstream multi- or unimodal tasks. We hope CLiMB will allow systematic evaluation of new models and algorithms for multimodal continual learning. CLiMB is designed to be an extensible community tool for studying tasks, model architectures, and CL algorithms.

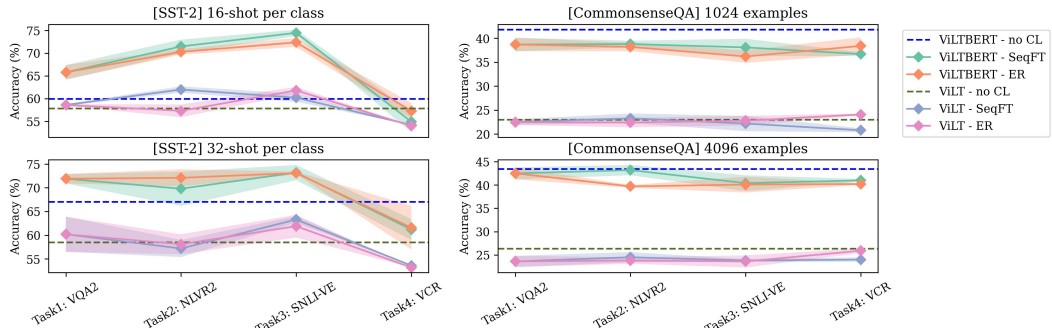

Figure 6: Comparisons between different encoders and continual learning algorithms on downstream language-only tasks (left: SST-2; right: CommonsenseQA). Note that the proposed Low-Shot Transfer metric is computed relative to the pre-trained encoder, making scores for different encoders (in this case, ViLT and VAuLT) incomparable. Hence, we plot the absolute accuracy with shaded standard deviation. VAuLT strictly improves absolute accuracy over ViLT in direct fine-tuning and CL settings. See Supp for comparable IMDb, HellaSwag, and PIQA results.

# 7    Limitations

**Task-Specificity**    The current CLiMB design allows for task-specific parameters and for model awareness of the task, but multi-task language modelling has seen impressive results from reframing all tasks as sequence-to-sequence problems that remove task-specific parameters [Raffel et al., 2020]. In future iterations of CLiMB, we intend to explore this task-agnostic paradigm, building further on the promising Adapters methods by learning a library of adapters that are dynamically selected based on input vision and language on a per-instance basis. Additionally, the task formulations in CLiMB are mostly classification, but sequence-based vision-and-language tasks could allow the study of embodied navigation [Anderson et al., 2018] and task completion [Shridhar et al., 2020], and may be feasible in a more task-agnostic CLiMB framework.

**Additional CL Metrics**    We have defined a set of metrics and methodologies for the challenge of multimodal continual learning, but these metrics are only an initial starting point. We design CLiMB to be flexible so that researchers can add metrics that they find valuable to measure, such as intransigence [Chaudhry et al., 2018].

**Ethical Considerations**    The initial CLiMB release is limited to English-only text, eliding the challenges of multi-lingual language tasks. Further, images in currently included datasets are sourced from social media, movie clips and web searches, thus excluding certain image domains, including those taken for accessibility needs such as descriptions for people with blindness [Gurari et al., 2018]. Such biases in a benchmark, inherited from the multi- and unimodal datasets selected, serve the needs of English-speaking, able-bodied folks as a "default."

# 8    Future Work

The initial findings from CLiMB reveal several promising opportunities and lines of research.

**Adapters**    Primarily, we find that Adapters are effective at mitigating catastrophic forgetting, while achieving comparable performance to full model fine-tuning. However, our current Adapter experiments introduce an independent set of parameters for each multimodal task, which does not facilitate sharing of task knowledge between tasks. Within unimodal multi-task and continual learning, Hypernetworks [Mahabadi et al., 2021] and compositional Adapter modules [Zhang et al., 2022] have been shown to facilitate knowledge transfer by generating Adapter parameters from shared task information. We plan to investigate how these methods generalize to multimodal CL, where shared information across tasks in either one or both modalities can influence generation of Adapter module parameters for new tasks.

**Distribution Shifts with Multiple Modalities**   Second, the stark performance degradation of the CL model after training on VCR, and the subsequent poor downstream few-shot transfer, invites study of how domain shifts in both vision and language inputs can affect upstream learning and forgetting, and can be mitigated.

**Sequence-to-Sequence Tasks**   Finally, as we noted in our Limitations, currently CLiMB only supports classification tasks. However, recently several "generalist" models have been developed, such as UnifiedIO [Lu et al., 2022] and FLAVA [Singh et al., 2022], that can solve a large variety of multimodal and unimodal tasks by formulating all tasks as a Sequence-to-Sequence problem. We plan to extend CLiMB to support such all-purpose Sequence-to-Sequence models.

## Acknowledgments and Disclosure of Funding

This work was supported by the Laboratory for Analytic Sciences (LAS), National Security & Special Research Initiatives, and in part by DARPA under contract HR001121C0168.

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
