# A  Multimodal Task Details

Table 4 shows details about individual multimodal tasks, including hyperparameters used to train ViLT for each task, and details about how low-shot versions of each task are sampled.

For NLVR2 and SNLI-VE, where the output labels are a small number of semantically meaningful categories (True/False and Entailment/Contradiction/Neutral respectively), we sample $N$ shots per output label to construct our low-shot training data. The 4 output labels in VCR are not semantically meaningful (since the options are interchangeable); hence, instead of sampling an equal number of training samples per label, we sample a percentage of the full training data instead. For VQAv2, the output label space is very large, and answers are not uniformly distributed across the training data, so instead of sampling $N$ shots per output label (answer) we again sample a percentage of the full VQAv2 training data.

| Task | VQAv2 | NLVR2 | SNLI-VE | VCR (Q $\rightarrow$ A) |
|---|---|---|---|---|
| | | Task Details | | |
| Task Type | Classification | Classification | Classification | Multi-Choice |
| Visual Input | 1 Image | 2 Images | 1 Image | 1 Image, Object boxes |
| Text Input | Question | Statement | Hypothesis | 1 question, 4 answers |
| # Output Labels | 3129 | 2 | 3 | 4 |
| Random Score, $S_R^i$ (%) | 0.0 | 50.0 | 33.33 | 25.0 |
| | | Training Details/Hyperparameters | | |
| Learning Rate | $10^{-4}$ | $10^{-4}$ | $5 \times 10^{-5}$ | $10^{-4}$ |
| Weight Decay | $10^{-2}$ | $10^{-2}$ | $10^{-2}$ | $10^{-2}$ |
| Adam Epsilon | $10^{-8}$ | $10^{-8}$ | $10^{-8}$ | $10^{-8}$ |
| Num. Epochs | 10 | 10 | 5 | 10 |
| Batch Size | 64 | 32 | 64 | 16 |
| | | Low-Shot Task Transformation | | |
| Number of shots per class | - | 2048 | 2048 | - |
| % of training data | 5% | 4.74% | 1.16% | 5% |

Table 4: Task-specific implementation details

# B  ViLT Model Modification Details

## B.1  Applying ViLT to Multi-Choice Tasks

### B.1.1  Applying ViLT to VCR

The VCR task provides object boxes, with each box corresponding to a grounded entity in the question. Unlike other pre-trained vision-language encoders [Su et al., 2019, Chen et al., 2020] that use visual features from regions-of-interest (ROIs) in the image, ViLT is designed to operate over image patches, thus making it challenging to use the object box inputs provided in the VCR task. We follow previous work [Zellers et al., 2021, Hessel et al., 2022] and draw colored boxes directly on the image corresponding to grounded references in the text. The grounded text references, *e.g.* [person1], [car1], are replaced with gender-neutral names for persons and object class names for all other objects. We use consistent mappings between the box colors and object names; for example, the [person1] object is always referenced with a green box in the image, and the name Casey in the text.

During training and inference, each possible answer $a_i$ is paired with the question $q$, to form a sequence "[CLS] $q$ [SEP] $a_i$". Each question-answer option is passed into the ViLT transformer, and the classifier produces a scalar score for each choice on top of the [CLS] representation. The choice with the maximum score is selected as the answer.

### B.1.2 Applying ViLT to HellaSwag, PIQA, and CommonsenseQA

The inputs of language-only multiple-choice tasks consist of two parts: a sentence $s$ (a sentence prefix in HellaSwag; a question in PIQA and CommonsenseQA), and a set of choices $A = \{a_1, a_2, ..., a_n\}$. We follow the original implementations [Zellers et al., 2019b, Bisk et al., 2020] to model these tasks, which consider different choices independently. For each choice $a_i$, we concatenate $s$ and $a_i$ with special tokens as the input: "[CLS] $s$ [SEP] $a_i$ [SEP]". We build the classifier, which outputs a scalar score for each choice, atop the [CLS] representation of ViLT transformer. During fine-tuning, we aggregate the scores of different choices and train the model with cross-entropy loss over the choices.

### B.2 Applying ViLT to Unimodal Tasks

**Sub-sampling.** We conduct low-shot experiments to test the model's transferability to unimodal tasks. However, different sub-samples the training set may lead to different results. To deal with this issue, for every language-only task, we use three different random seeds for sub-sampling, leading to three different training subsets, and then report the mean and standard deviation of the accuracy scores on the full validation set. For vision-only tasks, however, we observed low variances in accuracy across three sub-samplings ($39.35 \pm 0.4\%$ on Places 365; 16-shot per class). Thus, we fix the random seed and only use a single training subset for vision-only tasks due to the computational cost.

**What's the language input for vision-only tasks?** For vision-only tasks, we found that simply using "This is an image." as the language input works empirically well on all tasks. While the performance could potentially be further improved by using more informative and contextualized textual inputs, we leave it as future work as this is not the primary focus of this paper.

**What's the visual input for language-only tasks?** For language-only tasks, to keep the visual input in-distribution, we first resize all MS-COCO training images into $384 \times 384$ and then average them into a single image (see Fig 7) as the vacuous visual input, since MS-COCO is one of the pre-training corpora of ViLT. 384 is the shorter-edge image size used by ViLT, and with the default $32 \times 32$ patch projection, it takes $(384/32) \times (384/32) = 144$ image tokens in the original implementation. We also conduct ablation studies that include two baselines: (1) not inputting any image to ViLT at all, and (2) inputting the zero-vector image instead of the average image of the COCO dataset. The first three rows in Table 5 show that inputting the average image is slightly better than the other two baselines, presenting the benefit of using an in-distribution image, even when it is vacuous.

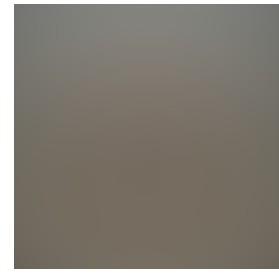

Figure 7: The average image of the MS-COCO dataset.

**Language-and-vision token reallocation.** In language-only tasks, we would like to focus on the language inputs, which only accounts for 40 tokens in the original ViLT implementation, instead of the vacuous visual input, which accounts for 144 tokens. Thus, we extend the language inputs by extending the positional embeddings to a maximum of 160, which is jointly learned during fine-tuning, and decrease the image tokens by downsample the image size to $128 \times 128$, which now only takes $(128/32) \times (128/32) = 16$ image tokens. The last row in Table 5 shows that this re-allocation of language-and-vision position embedding tokens notably improves the performance on language tasks.

|  | 16-shot | 32-shot | 128-shot |
|---|---|---|---|
| ViLT-40 -no image | $51.2 \pm 0.4$ | $54.0 \pm 1.2$ | $56.8 \pm 1.2$ |
| ViLT-40 -zero | $53.8 \pm 0.1$ | $54.8 \pm 0.3$ | $56.7 \pm 0.7$ |
| ViLT-40 -avg | $53.7 \pm 0.8$ | $55.1 \pm 1.0$ | $58.0 \pm 0.6$ |
| ViLT-160 -avg | $\mathbf{55.9} \pm 2.1$ | $\mathbf{57.8} \pm 1.5$ | $\mathbf{62.3} \pm 0.5$ |

Table 5: Accuracy (%) of vacuous visual input variants on IMDb, with $N = \{16, 32, 128\}$ shot per class. $-l$ means the maximum language sequence length is $l$, where ViLT-160 -avg is the proposed method that reallocates the language-and-vision tokens and has fewer visual tokens than other rows.

### B.3 VAuLT Implementation Details

The VAuLT model is a modification of the ViLT model that uses stronger language priors. Since the ViLT Transformer was initialized using weights from the vision transformer ViT [Dosovitskiy et al., 2020], and pre-trained only on image caption datasets, the language understanding of ViLT is limited to a specific language domain of image captions, thus making it unsuitable for language-only tasks. We perform additional experiments with a VAuLT model [Chochlakis et al., 2022] that replaces the language input embeddings of the ViLT Transformer with language token representations extracted from a frozen, pre-trained BERT model. VAuLT has more effective language understanding ability due receiving inputs from BERT, but the more general language representations could hurt its performance on vision-language tasks.

#### B.3.1 ViLT vs VAuLT Multimodal Task Comparison

Table 6 shows a comparison of pre-trained ViLT and VAuLT when directly trained on each of the upstream vision-language tasks. VAuLT underperforms ViLT across all multimodal tasks.

| Model | VQAv2 | NLVR2 | SNLI-VE | VCR |
|-------|-------|-------|---------|-----|
| ViLT | 67.70% | 73.07% | 76.31% | 61.31% |
| VAuLT | 65.80% | 65.57% | 74.12% | 59.46% |

Table 6: Comparison of pre-trained ViLT versus VAuLT when trained directly on each of the upstream multimodal tasks. VAuLT consistently underperforms ViLT's accuracy.

#### B.3.2 ViLT vs VAuLT Language-Only Task Comparison

Table 7 compares pre-trained ViLT and VAuLT when directly fine-tuned on downstream language-only tasks, showing that VAuLT can significantly improve the accuracy over ViLT.

| Model | IMDb | | SST-2 | |
|-------|------|------|-------|------|
| | 16 | 32 | 16 | 32 |
| ViLT | 55.9 ± 2.1 | 57.8 ± 1.5 | 57.8 ± 3.6 | 58.5 ± 7.4 |
| VAuLT | **64.8** ± 2.0 | **70.0** ± 1.7 | **59.9** ± 3.0 | **67.0** ± 3.2 |

| Model | HellaSwag | | CommonsenseQA | | PIQA | |
|-------|-----------|------|---------------|------|------|------|
| | 1024 | 4096 | 1024 | 4096 | 1024 | 4096 |
| ViLT | 26.5 ± 0.3 | 27.7 ± 0.3 | 23.0 ± 2.0 | 26.3 ± 0.5 | 52.0 ± 0.7 | 54.8 ± 0.5 |
| VAuLT | **31.7** ± 0.5 | **32.9** ± 0.8 | **41.8** ± 0.6 | **43.4** ± 0.6 | **54.6** ± 1.0 | **58.0** ± 0.8 |

Table 7: Comparisons between ViLT and VAuLT on downstream language-only tasks. Each value is the average accuracy (%) and standard deviation over 3 runs. We experiment with $\{16, 32\}$-shot per class on IMDB, SST-2 and sub-sample $\{1024, 4096\}$ training data for HellaSwag, CommonSenseQA, and PIQA. VAuLT consistently achieves higher accuracy than ViLT.

Similarly, Figure 8 shows that VAuLT strictly improves absolute accuracy over ViLT in direct fine-tuning and CL settings for downstream language-only tasks.

## C  Algorithm Implementation Details

For **Sequential Fine-tuning**, we fine-tune the shared encoder parameters when learning each task, whereas for **Frozen Encoder**, we keep the shared encoder frozen and only fine-tune the task-specific classification layers. In **Frozen Bottom-K**, the embedding parameters and the bottom $K$ ($< 12$) transformer layers are frozen; we set $K$=9 in our experiments.

The **Experience Replay (ER)** algorithm has two hyperparameters: the percentage of each task's training samples to be stored in the memory buffer, and the frequency with which to perform a "replay

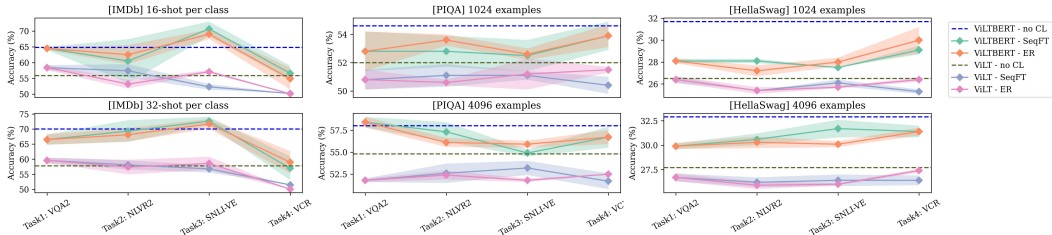

Figure 8: Comparisons between ViLT and VAuLT with checkpoints from different CL algorithms on downstream language-only tasks. We conduct three runs of different training sub-samplings and plot the absolute accuracy with shaded standard deviation.

step". We set these hyperparameters as 1% of training data and 100 training steps, respectively. We sample training examples for the memory buffer randomly from the training dataset; alternatives include sampling an equal number of training examples per output class.

**Elastic Weight Consolidation (EWC)** consists of computing the Fisher information matrix from the training data, which determines the importance of each parameter in the shared encoder. Rather than doing a full pass through the whole training data to construct the Fisher information matrix for each task, we use only 1% of the training examples. During training an upstream task, we sample one of the previous tasks and compute the L2 loss between parameter values in the current encoder and the previous task's encoder checkpoint. The L2 loss is weighted by that parameter's Fisher information and summed across all parameters. This EWC loss $\mathcal{L}_{EWC}$ is multiplied by a constant $\lambda$ and added to the upstream task loss $\mathcal{L}_{task}$. We select $\lambda = 10^2$ based on a hyperparameter sweep.

**Adapters** add a 2-layer MLP, also known as an Adapter module, after every Self-Attention and Feed-forward layer in each Transformer block. Following the original Houlsby configuration [Houlsby et al., 2019], the first layer of each Adapter module is a downsampling layer, which reduces the dimensionality of the input features by a factor of 16, followed by a GELU activation function, and finally an upsampling layer which produces an output representation with the same dimensionality as the Adapter input.

# D   Full ViLT Results

## D.1   Full Catastrophic Forgetting Results

In Table 8, we present full forgetting transfer numbers for all six CL algorithms, which were reported in a compact form in Figure 3a.

## D.2   Full Results of Different Upstream Task Orders

Table 9a contains full results of upstream knowledge transfer $\mathbb{T}_{UK}(i)$, when the ViLT encoder sees different sequences of upstream tasks. We use Sequential Fine-tuning for all these experiments. Table 9b shows the forgetting of previous tasks, for these different upstream task orders. We previously summarized these results visually in Figure 3b.

## D.3   Full Results of Low-Shot Multimodal Transfer

In Table 10, we present the full results when the CL-learned ViLT encoder, after training on the $i^{th}$ task, is trained on future low-shot tasks $\mathcal{T}_{VL}^{LS(j)}$ for $j = \{i + 1, ..., K_{VL}\}$. The first section of the table contains a comparison of ViLT's performance when directly fine-tuned on each task, when both full training data and low-shot versions of the task are available. The following sections show the low-shot transfer when upstream checkpoints, trained using four of our six CL algorithms, are used to fine-tuned on low-shot tasks. We do not perform experiments with the Frozen-Encoder and Adapter algorithms, as the encoder parameters are identical to the pre-trained checkpoint.

### CL Algorithm: Sequential Fine-tuning

| Evaluated on
Checkpoint | Task 1
VQAv2 | Task 2
NLVR2 | Task 3
SNLI-VE |
|---|---|---|---|
| After training on that task | [67.79] | [72.66] | [74.89] |
| Task 2: NLVR2 | 40.97% [40.02] | - | - |
| Task 3: SNLI-VE | 39.25% [41.18] | 43.81% [62.73] | - |
| Task 4: VCR | 63.90% [24.47] | 93.74% [51.24] | 89.93% [37.52] |

### CL Algorithm: Frozen Encoder

| Evaluated on
Checkpoint | Task 1
VQAv2 | Task 2
NLVR2 | Task 3
SNLI-VE |
|---|---|---|---|
| After training on that task | [58.15] | [63.66] | [69.45] |
| Task 2: NLVR2 | -0.38% [58.37] | - | - |
| Task 3: SNLI-VE | -0.38% [58.37] | -0.31% [63.70] | - |
| Task 4: VCR | -0.38% [58.37] | -0.42% [63.72] | 0.00% [69.45] |

### CL Algorithm: Frozen Bottom-9

| Evaluated on
Checkpoint | Task 1
VQAv2 | Task 2
NLVR2 | Task 3
SNLI-VE |
|---|---|---|---|
| After training on that task | [67.30] | [72.94] | [74.90] |
| Task 2: NLVR2 | 16.97% [55.90] | - | - |
| Task 3: SNLI-VE | 21.36% [52.93] | 29.32% [66.21] | - |
| Task 4: VCR | 71.61% [19.11] | 78.52% [54.93] | 35.01% [60.34] |

### CL Algorithm: Experience Replay

| Evaluated on
Checkpoint | Task 1
VQAv2 | Task 2
NLVR2 | Task 3
SNLI-VE |
|---|---|---|---|
| After training on that task | [67.87] | [73.20] | [75.08] |
| Task 2: NLVR2 | 12.88% [59.13] | - | - |
| Task 3: SNLI-VE | 12.96% [59.07] | 17.10% [69.23] | - |
| Task 4: VCR | 43.62% [38.27] | 78.27% [55.04] | 33.45% [61.11] |

### CL Algorithm: Elastic Weight Consolidation

| Evaluated on
Checkpoint | Task 1
VQAv2 | Task 2
NLVR2 | Task 3
SNLI-VE |
|---|---|---|---|
| After training on that task | [67.84] | [72.39] | [74.38] |
| Task 2: NLVR2 | 39.81% [40.83] | - | - |
| Task 3: SNLI-VE | 31.52% [46.46] | 25.73% [66.66] | - |
| Task 4: VCR | 65.25% [23.58] | 81.03% [54.25] | 73.61% [43.34] |

### CL Algorithm: Adapters

| Evaluated on
Checkpoint | Task 1
VQAv2 | Task 2
NLVR2 | Task 3
SNLI-VE |
|---|---|---|---|
| After training on that task | [68.10] | [73.66] | [76.08] |
| Task 2: NLVR2 | -0.01% [68.11] | - | - |
| Task 3: SNLI-VE | 0.04% [68.07] | 3.51% [72.83] | - |
| Task 4: VCR | 0.67% [67.64] | 6.48% [72.13] | 0.89% [75.70] |

Table 8: Full numbers for forgetting transfer $\mathbb{T}_F(j \leftarrow i)$ of previously seen tasks for each CL algorithm. We also show the transfer score $[S_{\mathcal{A}}^{j \leftarrow i}]$ when evaluated on that task after training on future task $i$. The first row contains task score $[S_{\mathcal{A}}^{j}]$ after originally training on $j^{th}$ task.

## D.4 Full Results of Low-Shot Unimodal Transfer

**Vision-only downstream tasks.** Table 11 presents the full results of vision-only tasks in absolute accuracy (%). Figure 9 plots the same results with Low-Shot Transfer (%). First, in single-task fine-tuning, we only include a single task in the upstream phase and compare the influence of different upstream tasks to downstream low-shot transfer. We found that across all vision-only downstream tasks, SNLI-VE > VQAv2 > NLVR2 > VCR, where VCR as the upstream task significantly damages the model performance. Second, current CL algorithms always *hurt* low-shot transfer compared to direct fine-tuning. Among them, Frozen Bottom-9 is the least harmful algorithm. Experience Replay

| Directly fine-tuning pre-trained ViLT on each task | | | |
|---|---|---|---|
| VQAv2 | NLVR2 | SNLI-VE | VCR |
| [67.70] | [73.07] | [76.31] | [61.31] |

| Task Order: VQAv2 → NLVR2 → SNLI-VE → VCR | | | |
|---|---|---|---|
| Task 1 | Task 2 | Task 3 | Task 4 |
| VQAv2 | NLVR2 | SNLI-VE | VCR |
| 0.13% [67.79] | -1.80% [72.66] | -3.33% [74.89] | -5.09% [59.47] |

| Task Order: SNLI-VE → VCR → VQAv2 → NLVR2 | | | |
|---|---|---|---|
| Task 1 | Task 2 | Task 3 | Task 4 |
| SNLI-VE | VCR | VQAv2 | NLVR2 |
| -0.07% [76.29] | -1.55% [60.75] | -6.55% [63.27] | -21.35% [67.65] |

| Task Order: NLVR2 → VQAv2 → VCR → SNLI-VE | | | |
|---|---|---|---|
| Task 1 | Task 2 | Task 3 | Task 4 |
| NLVR2 | VQAv2 | VCR | SNLI-VE |
| 0.06% [73.25] | -1.52% [66.55] | -6.03% [59.10] | -7.88% [73.07] |

(a) Full knowledge transfer results with different task orders.

| Task Order: VQAv2 → NLVR2 → SNLI-VE → VCR | | | |
|---|---|---|---|
| Evaluated on / Checkpoint | Task 1 VQAv2 | Task 2 NLVR2 | Task 3 SNLI-VE |
| After training on that task | [67.79] | [72.66] | [74.89] |
| Task 2: NLVR2 | 40.97% [40.02] | - | - |
| Task 3: SNLI-VE | 39.25% [41.18] | 43.81% [62.73] | - |
| Task 4: VCR | 63.90% [24.47] | 93.74% [51.24] | 89.93% [37.52] |

| Task Order: SNLI-VE → VCR → VQAv2 → NLVR2 | | | |
|---|---|---|---|
| Evaluated on / Checkpoint | Task 1 SNLI-VE | Task 2 VCR | Task 3 VQAv2 |
| After training on that task | [76.29] | [60.75] | [63.27] |
| Task 2: VCR | 84.50% [39.99] | - | - |
| Task 3: VQAv2 | 85.86% [39.40] | 91.47% [28.05] | - |
| Task 4: NLVR2 | 77.56% [42.97] | 86.11% [29.97] | 41.94% [36.73] |

| Task Order: NLVR2 → VQAv2 → VCR → SNLI-VE | | | |
|---|---|---|---|
| Evaluated on / Checkpoint | Task 1 NLVR2 | Task 2 VQAv2 | Task 3 VCR |
| After training on that task | [73.25] | [66.55] | [59.10] |
| Task 2: VQAv2 | 58.06% [59.68] | - | - |
| Task 3: VCR | 90.63% [52.16] | 68.69% [20.87] | - |
| Task 4: SNLI-VE | 91.75% [51.90] | 62.59% [24.94] | 34.04% [47.51] |

(b) Full forgetting results with different task orders.

Table 9: Effects of CL task order on the ViLT encoder's upstream knowledge transfer and forgetting.

and EWC perform similarly, and both are notably better than Sequential Fine-Tuning after training on VCR. In conclusion, the pre-trained ViLT already achieves decent performance on low-shot vision-only classification tasks. Meanwhile, with current CL algorithms, the model does not benefit from training on more vision-and-language upstream tasks, but suffers from forgetting useful visual representations, learned in pretraining, for downstream tasks.

**Language-only downstream tasks.** Table 12 presents the full results of language-only tasks in absolute accuracy (%). First, ViLT performs poorly on language-only tasks. Similar to our findings in vision-only tasks and multimodal tasks, including VCR as one of the upstream tasks hurts the model performance on downstream tasks, most notably on SST-2 and IMDb. Although VCR is also a multiple-choice commonsense reasoning task, it does not benefit HellaSwag, CommonsenseQA, and

| Directly fine-tuning on each task | | | |
|---|---|---|---|
| | Task 2 | Task 3 | Task 4 |
| Training Data presented | NLVR2 | SNLI-VE | VCR |
| Full Training Data, $S_{PT}^i$ | [73.07] | [76.31] | [61.31] |
| Low-Shot Transfer, $S_{PT}^{LS(i)}$ | [62.46] | [65.67] | [43.23] |

| CL Algorithm: Sequential Fine-tuning | | | |
|---|---|---|---|
| Low-Shot Transfer to | Task 2 | Task 3 | Task 4 |
| | NLVR2 | SNLI-VE | VCR |
| After training on Task 1: VQAv2 | -8.19% [61.44] | -4.51% [64.21] | -13.71% [40.73] |
| After training on Task 2: NLVR2 | - | -14.87% [60.86] | -26.60% [38.48] |
| After training on Task 3: SNLI-VE | - | - | -18.71% [39.82] |

| CL Algorithm: Frozen Bottom-9 | | | |
|---|---|---|---|
| Low-Shot Transfer to | Task 2 | Task 3 | Task 4 |
| | NLVR2 | SNLI-VE | VCR |
| After training on Task 1: VQAv2 | -15.73% [60.50] | -0.87% [65.39] | -4.22% [42.46] |
| After training on Task 2: NLVR2 | - | -0.99% [65.35] | -10.48% [41.32] |
| After training on Task 3: SNLI-VE | - | - | -4.00% [42.50] |

| CL Algorithm: Experience Replay | | | |
|---|---|---|---|
| Low-Shot Transfer to | Task 2 | Task 3 | Task 4 |
| | NLVR2 | SNLI-VE | VCR |
| After training on Task 1: VQAv2 | -7.95% [61.47] | -1.76% [65.10] | -15.47% [40.41] |
| After training on Task 2: NLVR2 | - | -10.48% [62.28] | -26.82% [38.34] |
| After training on Task 3: SNLI-VE | - | - | -18.38% [39.88] |

| CL Algorithm: Elastic Weight Consolidation | | | |
|---|---|---|---|
| Low-Shot Transfer to | Task 2 | Task 3 | Task 4 |
| | NLVR2 | SNLI-VE | VCR |
| After training on Task 1: VQAv2 | -13.24% [60.81] | -2.01% [65.02] | -15.52% [40.40] |
| After training on Task 2: NLVR2 | - | -17.53% [60.00] | -29.29% [37.89] |
| After training on Task 3: SNLI-VE | - | - | -22.87% [39.06] |

Table 10: Full low-shot multiodal transfer results, when transferring ViLT checkpoints from upstream CL training to future multimodal tasks.

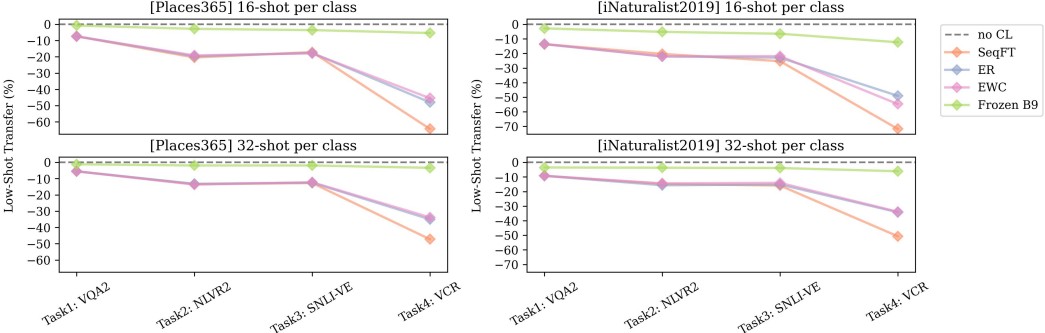

Figure 9: Low-Shot Transfer (%) comparison between different CL algorithms on downstream vision-only tasks (left: Places365; right: iNaturalist2019).

PIQA. On the other hand, continual learning sometimes improves the accuracy, especially on SST-2 and IMDb. CLiMB facilitates further investigation into these phenomena.

# E Experiments Using Another Vision-Language Model: UNITER

We conduct CL experiments using UNITER Chen et al. [2020] as the encoder. UNITER uses region features from a pre-trained Faster-RCNN as the visual input, in contrast to ViLT which directly

| Checkpoint \ Task | ImageNet 16 | ImageNet 32 | iNat2019 16 | iNat2019 32 | Places365 16 | Places365 32 | COCO 5% | COCO 10% |
|---|---|---|---|---|---|---|---|---|
| *Direct Fine-Tuning* | | | | | | | | |
| ViLT | 64.4 | 67.7 | 46.3 | 54.1 | 39.2 | 41.7 | 77.1 | 78.5 |
| *CL: Singe-Task Fine-Tuning* | | | | | | | | |
| After Task1: SNLI-VE | 62.3 | 66.3 | 43.6 | 53.1 | 37.6 | 40.5 | 74.6 | 77.4 |
| After Task1: VQAv2 | 58.8 | 63.3 | 40.0 | 49.1 | 36.3 | 39.4 | 73.2 | 75.7 |
| After Task1: NLVR2 | 56.2 | 62.7 | 36.4 | 48.4 | 31.9 | 37.0 | 67.3 | 73.1 |
| After Task1: VCR | 25.2 | 45.8 | 10.3 | 34.4 | 17.6 | 26.6 | 60.7 | 66.8 |
| *CL: Sequential Fine-Tuning* | | | | | | | | |
| After Task2: NLVR2 | 59.0 | 50.8 | 36.9 | 46.1 | 31.2 | 36.1 | 68.7 | 72.3 |
| After Task3: SNLI-VE | 51.5 | 59.1 | 34.6 | 45.5 | 32.5 | 36.4 | 70.3 | 72.6 |
| After Task4: VCR | 17.3 | 33.1 | 13.1 | 26.7 | 14.0 | 22.0 | 55.1 | 62.0 |
| *CL: Experience Replay* | | | | | | | | |
| After Task2: NLVR2 | 52.0 | 59.1 | 36.1 | 45.6 | 31.5 | 36.2 | 70.1 | 72.4 |
| After Task3: SNLI-VE | 52.2 | 58.8 | 35.7 | 45.9 | 32.3 | 36.5 | 70.6 | 72.9 |
| After Task4: VCR | 31.6 | 45.0 | 23.6 | 35.6 | 20.4 | 27.1 | 59.7 | 65.9 |
| *CL: EWC* | | | | | | | | |
| After Task2: NLVR2 | 52.6 | 59.6 | 36.1 | 46.3 | 31.7 | 36.0 | 69.5 | 72.6 |
| After Task3: SNLI-VE | 52.9 | 59.2 | 36.2 | 46.5 | 32.2 | 36.6 | 70.2 | 73.0 |
| After Task4: VCR | 30.4 | 45.0 | 21.0 | 35.8 | 21.4 | 27.6 | 60.4 | 65.6 |
| *CL: Frozen Bottom-9* | | | | | | | | |
| After Task1: VQAv2 | 62.8 | 66.3 | 45.0 | 52.2 | 38.9 | 41.2 | 76.6 | 78.1 |
| After Task2: NLVR2 | 62.2 | 66.0 | 43.9 | 52.1 | 38.1 | 40.9 | 76.1 | 78.1 |
| After Task3: SNLI-VE | 62.0 | 65.8 | 43.3 | 52.0 | 37.8 | 40.9 | 75.9 | 77.9 |
| After Task4: VCR | 60.2 | 65.1 | 40.6 | 50.8 | 37.1 | 40.3 | 75.4 | 77.8 |

Table 11: Comparisons between different CL algorithms on vision-only tasks. We experiment with $\{16, 32\}$-shot per class on ImageNet-1000, iNaturalist 2019, and Places 365 datasets. For COCO multi-label object detection task, we sub-sample $\{5\%, 10\%\}$ of the training data. All CL algorithms hurt the accuracy (%) compared to direct fine-tuning, while Frozen Bottom-9 is the least harmful one. Comparing different upstream tasks, SNLI-VE > VQAv2 > NLVR2 > VCR across all four downstream tasks, where VCR greatly damages the performance. Note that for Sequential Fine-Tuning, Experience Replay, and EWC, the result of "After Task1: VQAv2" is shown under Single-Task Fine-Tuning, as there are no differences between these CL algorithms in the first task.

operates on image patch tokens. We train UNITER on the same sequence of four upstream tasks (VQA → NLVR2 → SNLI-VE → VCR), using all of our CL algorithms except Adapters.

In Table 13, we compare Upstream Knowledge Transfer between various CL algorithms, trained using the UNITER model. We see that UNITER, similar to ViLT, has negative transfer for post-VQA tasks, although UNITER typically has less negative transfer for the third task (SNLI-VE) than the second task (NLVR2).

In Figure 10, we plot Forgetting of previous tasks when the UNITER model is continually learned. We observe that Forgetting trends between algorithms are similar to our findings with ViLT: fine-tuning fewer shared parameters leads to less forgetting, while Experience Replay performs best among the algorithms that fine-tune all the shared parameters. In contrast to ViLT, we see that the VCR task does not impact the UNITER model's Forgetting as severely. This is likely due to the fact that UNITER (which utilizes region features) directly uses the ground-truth bounding boxes from the VCR task as part of the model, rather than drawing on the boxes onto the image as the image patch-based ViLT model does.

| Task / Checkpoint | IMDb | | SST-2 | | HellaSwag | | ComQA | | PIQA | |
|---|---|---|---|---|---|---|---|---|---|---|
| | 16 | 32 | 16 | 32 | 1024 | 4096 | 1024 | 4096 | 1024 | 4096 |
| *Direct Fine-Tuning* | | | | | | | | | | |
| ViLT | 55.9 | 57.8 | 57.8 | 58.5 | 26.5 | 27.7 | 23.0 | 26.3 | 52.0 | 54.8 |
| *CL: Singe-Task Fine-Tuning* | | | | | | | | | | |
| After Task1: SNLI-VE | 59.0 | 60.5 | 59.8 | 60.6 | 26.8 | 27.3 | 23.3 | 26.5 | 52.7 | 55.2 |
| After Task1: VQAv2 | 58.4 | 59.6 | 58.6 | 60.2 | 26.4 | 26.7 | 22.5 | 23.6 | 50.8 | 51.8 |
| After Task1: NLVR2 | 56.2 | 56.5 | 55.8 | 58.2 | 26.0 | 27.1 | 21.5 | 25.4 | 52.8 | 54.1 |
| After Task1: VCR | 49.9 | 51.1 | 51.3 | 51.5 | 26.6 | 26.7 | 21.9 | 24.3 | 49.3 | 51.9 |
| *CL: Sequential Fine-Tuning* | | | | | | | | | | |
| After Task2: NLVR2 | 57.4 | 58.1 | 62.0 | 57.2 | 25.4 | 26.2 | 23.3 | 24.5 | 51.1 | 52.6 |
| After Task3: SNLI-VE | 52.3 | 56.8 | 60.2 | 63.3 | 26.1 | 26.4 | 22.2 | 23.8 | 51.1 | 53.2 |
| After Task4: VCR | 50.2 | 51.5 | 54.2 | 53.6 | 25.3 | 26.4 | 20.8 | 24.0 | 50.4 | 51.7 |
| *CL: Experience Replay* | | | | | | | | | | |
| After Task2: NLVR2 | 53.2 | 57.4 | 57.3 | 58.1 | 25.4 | 25.9 | 22.4 | 23.8 | 50.6 | 52.4 |
| After Task3: SNLI-VE | 57.1 | 58.6 | 61.8 | 61.9 | 25.7 | 26.0 | 22.8 | 23.6 | 51.2 | 51.8 |
| After Task4: VCR | 50.1 | 50.1 | 54.0 | 53.2 | 26.4 | 27.4 | 24.1 | 25.9 | 51.5 | 52.5 |
| *CL: EWC* | | | | | | | | | | |
| After Task2: NLVR2 | 51.0 | 55.1 | 59.7 | 57.0 | 25.2 | 26.4 | 21.6 | 24.1 | 50.0 | 52.1 |
| After Task3: SNLI-VE | 53.9 | 54.5 | 57.1 | 57.5 | 25.7 | 26.1 | 20.6 | 22.4 | 50.0 | 52.8 |
| After Task4: VCR | 49.7 | 49.8 | 51.3 | 51.1 | 25.9 | 27.0 | 22.0 | 23.8 | 51.5 | 52.1 |
| *CL: Frozen Bottom-9* | | | | | | | | | | |
| After Task1: VQAv2 | 57.4 | 57.9 | 56.5 | 58.9 | 25.8 | 26.9 | 22.5 | 27.3 | 50.3 | 54.4 |
| After Task2: NLVR2 | 53.9 | 55.7 | 56.2 | 58.7 | 26.6 | 27.2 | 24.8 | 27.0 | 51.3 | 54.0 |
| After Task3: SNLI-VE | 57.8 | 61.6 | 56.8 | 60.7 | 25.7 | 27.1 | 23.5 | 26.6 | 51.1 | 53.0 |
| After Task4: VCR | 54.2 | 55.7 | 56.5 | 58.2 | 26.1 | 27.2 | 24.3 | 27.4 | 51.5 | 53.6 |
| Random | 50.0 | 50.0 | 50.0 | 50.0 | 25.0 | 25.0 | 20.0 | 20.0 | 50.0 | 50.0 |

Table 12: Comparisons between different CL algorithms on language-only tasks. Each value is the average accuracy (%) over 3 runs. We experiment with $\{16, 32\}$-shot per class on IMDB, SST-2. For HellaSwag, CommonsenseQA, and PIQA, we sub-sample $\{1024, 4096\}$ instances of training data.

| Alg $\mathcal{A}$ | Task 1 VQAv2 | Task 2 NLVR2 | Task 3 SNLI-VE | Task 4 VCR |
|---|---|---|---|---|
| Direct FT | [69.30] | [75.25] | [78.09] | [69.97] |
| SeqFT | 0.06% [69.34] | -4.76% [74.05] | -3.69% [76.44] | -11.92% [64.61] |
| Frozen Enc | -22.67% [53.59] | -58.82% [60.40] | -24.23% [67.24] | -48.85% [48.01] |
| Frozen B9 | -0.89% [69.91] | -5.16% [73.95] | -0.68% [77.79] | -9.80% [65.57] |
| ER | 0.06% [69.34] | -3.35% [74.41] | -2.26% [77.07] | -11.24% [64.92] |
| EWC | 0.06% [69.34] | -4.71% [74.06] | -3.52% [76.51] | -12.41% [64.39] |

Table 13: Upstream Knowledge Transfer $\mathbb{T}_{UK}(i)$ relative to direct fine-tuning on each task, along with task score $[S_{\mathcal{A}}^i]$ (%), for different CL algorithms $\mathcal{A}$ applied to UNITER.

# F  Hardware and Resources Used

Our experiments were performed on an Exxact workstation containing four NVIDIA RTX 3090 GPUs. Each upstream continual learning experiment was run on a single GPU. While individual tasks took between 12 hours and 2 days to train, the entire 4-task continual learning typically took between 4 and 5 days. For downstream experiments, each (upstream checkpoint, downstream task, sample size, random seed) run took between 30 minutes to 3 hours to train, depending on the tasks.

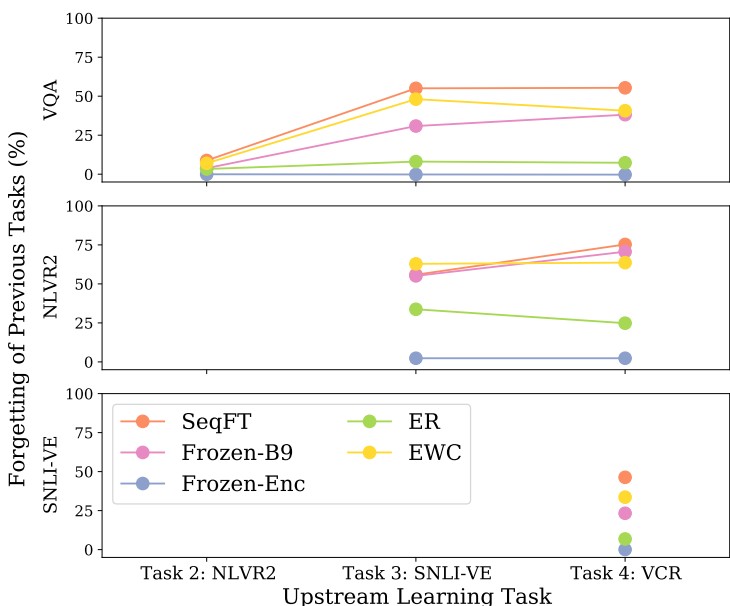

Figure 10: Forgetting $\mathbb{T}_F(j \leftarrow i)$ (%) of the previous $i-1$ tasks when the UNITER model is trained using each algorithm. Each subplot denotes model performance on one of the previous tasks. Similar to our findings with ViLT, ER best preserves past task performance among all algorithms that fine-tune shared parameters.