# OpenReview forum: "CLiMB: A Continual Learning Benchmark for Vision-and-Language Tasks"
_NeurIPS.cc/2022/Track/Datasets_and_Benchmarks — NeurIPS 2022 Datasets and Benchmarks _

### Official Review · Reviewer_hZLm · 2022-07-03
**An important challenge - good paper**

**Rating:** 7
**Confidence:** 4

**Strengths:**

“Continually learning” or adaptation is an important and timely challenge, especially in the rise of more tasks and models.
Extensive evaluation of tasks (multimodal, visual, textual) and CL algorithms.
Motivation new research of continual learning strategies for vision-language tasks.
An extensible framework.


**Weaknesses:**

The main weakness is that they evaluate only one model (ViLT), and the insights could be different when using other models.
Another weakness is pointing only to the problem without hinting at solutions.
I do think that this is a good paper. However, dealing with these 2 weaknesses can make it great.

There could also be a better evaluation of the order of the tasks, but I do understand that doing so is limited since there are many options and permutations of tasks.


**Additional Feedback:**

I hope that the benchmark can indeed be easily extended.
This is an important challenge. Identifying a problem is important, but I hope that your framework will pave a path to better solutions.


**Clarity:**

The paper is well written. I could use a few more examples to better understand the formulations in S3.1, perhaps in the Appendix.
Additionally I found some of the plots difficult to understand with the formulations, perhaps it will be helpful to explain a single point (e.g., Fig 4)


**Correctness:**

See the main weakness from the previous comment: they evaluate only one model (ViLT), and the insights could be different when using other models.

**Documentation:**

A GitHub repository is available.


**Ethics:**

I don’t find ethical concerns in this paper.


**Relation To Prior Work:**

Not enough. They call the subject “continual learning”, but literature also refers to it as “online learning” or “domain adaptation”, and there are many more previous works and strategies to consider.

**Summary And Contributions:**

A benchmark to study continually learning in multimodal tasks.
They show that CL strategies do not generalize well when trained on a sequence of multimodal tasks, or enable few-shot transfer.
Designed to be extensible for new tasks, architectures, and CL algorithms.

---

> ### Author Response · Authors · 2022-08-15
> **Response to Reviewer hZLm**
>
> We thank the reviewer for their comments and give some detailed responses below in addition to the General Response.
>
> **More than one vision-language model:** We refer the reviewer to General Response #1. We are also including UNITER into the CLiMB benchmark, and our revision contains preliminary CL experiments with the UNITER model in the supplementary. The camera-ready version of our paper will contain full UNITER experiments, including a comparison between the CL trends in our ViLT and UNITER experiments.
>
> **Pointing to the Problem:** The reviewer points out that our paper currently only points at the challenges involved in multimodal continual learning, without solving said challenges. Our intent in introducing the CLiMB benchmark in this paper was to introduce and motivate the problem of multimodal CL and provide researchers with a framework to investigate solutions to these challenges. Additionally, our findings from the initial CLiMB experiments indicate that Adapters can mitigate forgetting while achieving similar performance to full model finetuning methods. In future work, we plan to investigate how Adapter methods can be used to facilitate cross-task knowledge transfer in the multimodal setting, inspired by results in [1].
>
> **Extensibility:** We design CLiMB to be easily extensible to new tasks, models and CL algorithms. We refer the reviewer to the new guides added in our GitHub repository, that provide instructions on how to extend the benchmark to include new tasks, models and algorithms.
>
> [1] *Continual Sequence Generation with Adaptive Compositional Modules* Yanzhe Zhang, Xuezhi Wang, Diyi Yang, ACL 2022

---

### Official Review · Reviewer_sJXK · 2022-07-25
**Great insights into CL challenges but a poorly documented benchmark**

**Rating:** 4
**Confidence:** 5

**Strengths:**

- The paper tackles understudied problems, namely multi-modal CL and knowledge transfer from bi-modal to uni-modal continual learning.
- The discoveries are of high importance to the field.


**Weaknesses:**

The paper proposes an important benchmark to help researchers perform CL on bi- and uni-modal problems. However, this benchmark is poorly documented. I could identify these drawbacks:
- The implemented code misses any type of documentation: missing Docstrings, typing.
- The only thing that explains the benchmark is a readme file in the repository. There should be an illustrative tutorial showing how to perform and evaluate each setting.
- There should be a tutorial that explains how the different classes can be instantiated and used.
- There is no class or function diagram to show how the whole framework is constructed.
- There is no explanation of how the framework can be extended using a new CL method.
- There is no explanation of how to incorporate a new multi- or uni-modal data set.
- No maintenance plan.
- To be useful, the benchmark should facilitate a clear and easy way of creating reports that consist of plots and tables.




**Additional Feedback:**

- At the end of Page 3, why do you compare with the performance of S_{PT}^{LS(i)}? Shouldn't the comparison be against the task T^i_M itself?


**Clarity:**

The details of the proposed performance measures are put in Table 2, which is fine. However, they should be explained better. Is random selection the same as a random classifier?


**Correctness:**

The paper and the benchmark are limited to very few measures presented in the paper. The authors should take into account the measures proposed in Chaudhryet al.
Chaudhry, Arslan, et al. "Riemannian walk for incremental learning: Understanding forgetting and intransigence." Proceedings of the European Conference on Computer Vision (ECCV). 2018.


**Documentation:**

Poorly documented. See the weaknesses points.


**Ethics:**

Not relevant.


**Relation To Prior Work:**

Missing relation to former CL benchmarks.


**Summary And Contributions:**

The paper describes a benchmark to assess catastrophic forgetting of various CL methods on multi-modal data sets. It also studies how knowledge transfer is present when shifting to unimodal problems or to low-shot tasks.
The discoveries are very important and show how CL is still an unsolved problem. It's also interesting to see that adapters can mitigate forgetting while freezing the encoder part.

---

> ### Author Response · Authors · 2022-08-15
> **Response to Reviewer sJXK**
>
> We thank the reviewer for their feedback and suggestions, and for recognizing the importance of the understudied multimodal CL problem.
>
> **Documentation:** The reviewer makes great suggestions about improvements in documentation to our benchmark that are necessary to facilitate ease of use for other researchers who want to work on these problems. Some of these documentation efforts were in progress during the review period, but the reviewer also correctly highlights several components that were missing. To that end, we have updated the CLiMB GitHub repository (https://github.com/GLAMOR-USC/CLiMB) with the following documentations:
>
> - Docstrings and typing for all classes and methods required for CL:
>     - for data handling, in src/data/
>     - for model definition, in src/modeling/
>     - For task training, in src/train/visionlanguage_tasks/
>     - training scripts, in src/train/train_*.py
>     - CL evaluation, in src/cl_evaluation/evaluate_cl_algorithm.py, and
>     - CL algorithms, in src/cl_algorithms
> - Instructions on:
>     - how to add new multimodal tasks to CLiMB (ADD_NEW_TASKS.md)
>     - how to add new models (ADD_NEW_MODELS.md)
>     - How to add new algorithms (ADD_NEW_ALGORITHMSmd)
> - The above documents contain a diagram showing how the classes and methods are organized, and detailed information about each necessary class and method that needs to be defined, along with links to example definitions.
> - We also include a document (TRAIN_UPSTREAM_CL.md), with instructions on how to run the upstream CL setting and a flowchart demonstrating the CL process.
>
> **Creating Reports and Plots:** We currently provide scripts to calculate and save results for individual CL experiments (in src/cl_evaluation/). We will release a full plotting and reporting suite alongside the full UNITER results with the camera-ready paper, as we are developing it to be flexible enough to enable comparisons across models as well as algorithms.
>
> **Limited Metrics:** We thank the reviewer for pointing us to the “Understanding forgetting and intransigence” paper. While we do have a forgetting metric, we believe the intransigence metric (defined in that paper as “inability of an algorithm to learn new tasks”) is captured by our Upstream Knowledge Transfer metric, as negative Knowledge Transfer (i.e. worse performance on task in CL setting compared to single-task finetuning) measures the inability of the model to learn new tasks in a CL setting (the difference between our definitions is that they compare their CL performance to multi-task performance, whereas we compare to single-task performance). We have added a line on Page 6 making this connection between Upstream Knowledge Transfer and Intransigence in our revision.

---

### Official Review · Reviewer_iGvT · 2022-07-26
**CLiMB: A Continual Learning Benchmark for Vision-and-Language Tasks**

**Rating:** 6
**Confidence:** 4
**Clarity:** Good.

**Strengths:**

Clarity: This paper was clear and easy to follow for me.

Quality: Experiments were well done; the paper includes a variety of continual learning methods and presents a variety of different analyses.  The results lead to some interesting insights as well.  For example, L232 mentions that experiments indicate that current language and vision models do not learn complex language reasoning skills.  This makes sense given the data they are trained on, but it is nice to see quantitatively.  I appreciate that the paper includes many different continual learning algorithms.

Originality:  Continual learning is well studied in machine learning.  The originality in this paper comes from crafting a multimodal continual learning benchmark and providing analysis on common continual learning algorithms in a new domain.

Significance:  Continual learning is an important research area and creating a benchmark for multimodal continual learning will likely spur interesting research in this direction.


**Weaknesses:**

Clarity: One small thing I did not understand is how to read Figure 3b; it was not clear to me which lines correspond to which task order and how to draw conclusions from the chart.

Quality: The paper could be stronger by applying continual learning algorithms to more than one language and vision model (ViLT is used in the paper).  However, I do not think this is necessary for acceptance.

Originality: As one of the main contributions of this paper is presenting a continual learning multimodal task set up, I would have liked to have seen more discussion on why the task set up is a good one, perhaps by motivating with a “real world” example.  Whereas in the continual learning setup I am the most familiar with (adding new class labels sequentially) has an intuitive motivation that a model might come across new classes continuously, I was less sure of the motivation for adding new tasks sequentially.  Furthermore, though the author’s mention that unimodal tasks can benefit from multimodal tasks, the reverse can also be true (e.g,. many of our models include pretrained detectors).  Another way to set up multimodal continual learning would be to see if learning additional concepts in vision only or language only tasks transfers to tasks like VQA or NLVR (similar to what was done in some older work on “novel object captioning” see [here](https://arxiv.org/abs/1511.05284)).   I would like to see more explanation motivating the particular setup chosen in this paper.

Significance:  Though the paper is likely to spur more interest in multimodal continual learning, I am not completely sold on what is measured (e.g., upstream to downstream task transfer) and would be a bit concerned that others might adopt the dataset without questioning decisions made in this paper.


**Additional Feedback:**

n/a

**Correctness:**

As far as I can tell, the paper is correct.


**Documentation:**

Well documented; I checked out the github and though I didn’t download anything myself it looks fairly straightforward.

**Ethics:**

Addressed in discussion and limitations.

**Relation To Prior Work:**

Good – authors mention papers that show multimodal work can benefit unimodal tasks; another paper that could be cited is the [vokenization paper](https://arxiv.org/pdf/2010.06775.pdf).

**Summary And Contributions:**

This paper combines existing datasets (five language and vision datasets, five vision datasets, and five language datasets) to create a continual learning benchmark.  The paper then evaluates a standard language vision transformer model and six continual learning algorithms (including simple algorithms like sequential fine tuning and more complex algorithms such as including task adapters).  The paper then analyzes whether knowledge transfers between language and vision tasks, whether tasks are forgotten when updating models for new tasks, and whether models exhibit transfer to language and vision only tasks.

The main contributions are: continual learning benchmark for multimodal tasks and analysis on a variety of continual learning algorithms in the multimodal setting.

---

> ### Author Response · Authors · 2022-08-15
> **Response to Reviewer iGvT**
>
> We thank the reviewer for their comments. We would like to address the main weaknesses that the reviewer pointed out:
>
> **Figure 3b:** In both of the plots in Figure 3b, every colored shape corresponds to a different multimodal task (keyed by the legend), and every black line connecting a set of shapes left to right corresponds to a single task order (for e.g., green triangle -> pink diamond -> orange star -> teal circle corresponds to VQA->NLVR->SNLI-VE->VCR).
>
> In the Upstream Forward Transfer plot (left), tasks after VCR (teal circle) experience negative transfer: e.g. when task 2 is VCR (teal circle), the 3rd task VQA (green triangle) and 4th task NLVR (pink diamond) experience significantly more negative transfer than in other task orders.
>
> In the Forgetting plot (right), the average forgetting of previous tasks after learning VCR (teal diamond) is much higher (>=80%) than the average forgetting after learning the tasks before it. For e.g. when task 2 is NLVR (pink diamond) the forgetting of task 1 is 40%, then after learning the 3rd task SNLI-VE (orange star), the average forgetting of task 1 and 2 is still 40%, but after learning the 4th task VCR (teal diamond), the average forgetting of previous tasks jumps up to >80%.
>
> **More than one vision-language model:** We refer the reviewer to the general response #1 - we are also including UNITER into the CLiMB benchmark, and our revision contains preliminary CL experiments with the UNITER model. The camera-ready version of our paper will contain full UNITER experiments, including a comparison of how the CL trends in our ViLT experiments compare against those seen in UNITER.
>
> **Why is Task-Shifting CL Interesting?** The reviewer raises an important distinction between our work and the majority of CL literature that focuses on continually adapting to new classes. While we agree that label shift is also an important CL setting for real-world deployment of models, we envision a slightly different application of Machine Learning models. As we seek to deploy real-world agents (such as robots or chatbots) that can simultaneously solve a large number of tasks (as in multi-task learning) by perceiving and interacting with the real world, we also want these agents to learn new capabilities and functions continually. For instance, a multimodal dialog agent such as BlenderBot needs to continually add capabilities related to information retrieval, question answering, and multimodal dialog. As the community has begun to tackle tasks that include multiple input modalities, the CLiMB benchmark provides researchers with a benchmark where this ability of multimodal models to learn continually can be systematically studied.
>
> **Significance:** This is a really good point. We design the CLiMB benchmark to be easily extensible for other researchers, so they can add more datasets, models, and metrics (following the guides provided in our GitHub repository). As such, the metrics and methodologies in CLiMB are an initial starting point for the challenge of multimodal continual learning, but the benchmark is designed to be flexible so that researchers can decide for themselves what metrics and properties are valuable to measure. We include a discussion on it in our limitations.

---

### Official Review · Reviewer_M1cy · 2022-07-27
**CLiMB Review**

**Rating:** 7
**Confidence:** 5
**Correctness:** Yes
**Clarity:** Yes

**Strengths:**

The authors do a great job at defending their thesis. I was pleasantly surprised by a number of the findings, including the relatively higher performance of the Adapter method in mitigating forgetting over the Freezing Bottom 9 Layers method. This is definitely a finding that will get a lot of attention and conversations started on really digging into why this could be. As such, the paper lays a great foundation to future research into the use of Adapters as a preferred method to improve CL tasks.

**Weaknesses:**

There are no obvious weaknesses so I will simply share some of my "Would Have Been Nice Haves".
They go down particular strategies and the performance of said strategies on downstream task performance showing that all methods lead to a negative knowledge transfer capability except one: Adapters. This is where my interest get particularly peaked. Adapters not only are more performant than other forgetful mitigation strategies explored in this paper, but they also perform comparably to full model tuning. As well, there is a moment in the paper where the authors contend that model forgetfulness is also seemingly related to the sequence by which the models are being trained on each task. They show that one model in particular, VCR, leads to the most extreme level of forgetting and negatively affects knowledge transfer. I wish the authors would have spent more time on that. They only spend 1 sentence speculating on why this may be and I was left wanting more. Same for the Adapters observation, they quickly overlook the higher performance in the breakdown of the paper's results.
This brings me to another thing, ViLT-BERT. The authors do not discuss ViLT-BERT specifically until page 6. Before this, they focus the crux of their analysis around downstream task performance on how well the CL models can generalize to ViLT. This is fine, but I wish the authors did not bury the lede there. ViLT-BERT outperforms the low shot accuracy of ViLT which is extremely significant because it shows what the authors correctly point out in the paper: strong priors assist with downstream low shot performance. Considering the lengths the authors went into abstracting CL tasks into two phases -upstream and downstream- it would have been nice to see them speculate the relationship between upstream task performance and downstream task accuracy on both ViLT and ViLT-BERT keeping the idea of strong priors in mind throughout more of the paper.

**Additional Feedback:**

No additional feedback.

**Documentation:**

Yes, their Github repo is quite clean with a good Readme doc and clear expectations on how people should use the benchmark in future research.

**Relation To Prior Work:**

The authors do a good job at presenting prior contributions while showing a need for the addition of their work to present research.

**Summary And Contributions:**

The paper starts off with a simple preposition: it would serve as a benchmark to gauge the effectiveness of knowledge transfer and mitigated forgetfulness in cross-task modeling using both multi-modal and unimodal frameworks. They abstract these tasks into two phases: phase 1, upstream continual learning tasks, which considers the degree of forgetfulness and knowledge transfer across language and vision tasks trained sequentially, and phase 2, downstream continual learning tasks, which considers performance of low shot transfer capability of the models used in the prior phase.

Through the benchmark, the authors conclude that in the upstream phase, CL models do not lead to knowledge transfer AND that while some tasks mitigate forgetting, model performance pails in comparison to a full tuning of a model to a specific task. There are a couple interesting caveats to that, though: using Adapters is a strategy that the authors found outperforms other methods for forgetting mitigation and VCR is one task that greatly degrades performance on forgetting and knowledge transfer. They are able to show that current CL models are not able to appropriately generalize to tasks or be particularly performant for downstream low shot tasks.

---

> ### Author Response · Authors · 2022-08-15
> **Response to Reviewer M1cy**
>
> We thank the reviewer for their comments, and for their enthusiasm for some of our findings. In fact, the reviewer is excited by the same opportunities that we are seeking to further investigate using the CLiMB benchmark.
>
> **Adapters:** As the reviewer points out, Adapters perform comparably to full model tuning while mitigating forgetting. However, introducing separate parameters for each task prevents any knowledge transfer between tasks. In ongoing work, we are exploring techniques such as Hyperparameters [1] and compositional Adapter modules [2], which share knowledge between task parameters, and explore how they generalize to a multimodal CL setting.
>
> **Extreme Forgetting from VCR:** This result, as the reviewer correctly points out, is indeed striking, and invites further investigation into how domain shifts in both language and visual inputs affect continual learning of new tasks, and forgetting of already-learned tasks. Our findings with VCR show that task-specific tricks (such as drawing bounding boxes onto images) that can facilitate effective learning on individual tasks do not generalize well to a continual learning setting. Models such as UNITER, which do not use similar tricks because they utilize visual object features rather than image patches, might not show the same VCR-related issue that ViLT does.
>
>
> [1] *Parameter-efficient multi-task fine-tuning for transformers via shared hypernetworks*, Rabeeh Karimi Mahabadi, Sebastian Ruder, Mostafa Dehghani, and James Henderson, ACL 2021
>
> [2] *Continual sequence generation with adaptive compositional modules* Yanzhe Zhang, Xuezhi Wang, and Diyi Yang, ACL 2022

---

### Official Review · Reviewer_fn9P · 2022-07-28
**It identifies several problems in the multimodal continue learning setting, but needs more justifications on its significance.**

**Rating:** 6
**Confidence:** 4
**Correctness:** The experiments design are appropriat…
**Clarity:** The paper is well written and is easy…

**Strengths:**

- The paper identifies an interesting problem on the continual learning with vision-language pretraining, and constructs different data configuration settings to study this problem.
- A unified toolkit is built for studying the proposed task.

**Weaknesses:**

- The paper proposes a multimodal continual learning benchmark, evaluates the baseline approaches and identifies several issues with the current approaches.  However, it fails to explain why the identified problem is valuable for the community to solve: why do we need to tackle this problem specifically in the multimodal settings?  If it is specific to multimodal data (but not to unimodal data), the authors need to explain the status quo for the unimodal data.  If not, the authors need to analyze why it is better to tackle the problem in multimodal settings instead of unimodal settings.
    - The paper calls for future work exploring cross-task knowledge transfer for vision-language models (L11-13), without much of the evidence why this line of work is promising.  For example, is there any line of work in vision-only / language-only showing that different tasks in continual learning are complimentary.  If so, this would make the argument stronger; and if not, the authors need to explain why this would be promising in vision language — after combing the vision and language modality.
    - The author explains the performance drop of the VCR dataset with the large distribution shift between upstream and downstream tasks.  This seems to be a general problem with the continual learning.  Similarly, is there any evidence in any modality suggesting that this is solvable, or is there any arguments on why this can be easier with multimodal data?

**Additional Feedback:**

See weakness section above.  I would be willing to increase the rating if the above concerns are addressed.

**Documentation:**

The code is hosted on GitHub with documentations on the installation, data download, and how to train the model.

**Ethics:**

I do not see a significant concern on further discussion for ethical issues.

**Relation To Prior Work:**

The author has discussed the related works.

**Summary And Contributions:**

The paper identifies that the current CL benchmarks limit to the vision-only or language-only tasks, and there lacks a benchmark for vision-language models.  It proposes CLiMB, a benchmark that contains upstream vision-language pretraining and downstream multimodal tasks.  The paper compares and evaluates different CL approaches with ViLT model, and find that common CL methods do help mitigate catastrophic forgetting during multitask pretraining, but they are not complimentary.

---

> ### Author Response · Authors · 2022-08-15
> **Response to Reviewer fn9P**
>
> We thank the reviewer for their comments. We would like to address the three main weaknesses that the reviewer pointed out:
>
> **Why Multimodal CL is Important:** We refer the reviewer to General Response #2. Particularly, with respect to why we need to tackle the problem specifically in multimodal settings, we envision the deployment of AI agents into the real world that should be able to continually learn new tasks and functions utilizing multiple input modalities. As a first step towards building models that can learn new multimodal capabilities continually, we propose the CLiMB benchmark to study how multimodal tasks can be learned in a CL setting.
>
> **Evidence of Cross-Task Knowledge Transfer:** In NLP, it has been seen that knowledge transfer can occur across tasks in a continual learning setting, resulting in improvements over single-task finetuning [1]. Further, among multimodal tasks, it has been shown that knowledge can be transferred across tasks via multi-task learning. In the multimodal setting, VL-BERT trained via multitask learning [2] sees significant gains from cross-task knowledge transfer; in particular, the authors find that “a single model can perform all these tasks, but also that joint training can improve the performance compared to single-task training with the same architecture”. We believe that this ability to transfer knowledge across tasks should ideally be reflected in the continual learning setting as well, where models can utilize knowledge from already-learned tasks in order to better learn newly arrived tasks.
>
> **Evidence that Distribution Shift is Solvable in CL:** We agree that the challenge of distribution shift is particularly pronounced during continual learning and downstream transfer. Rather than framing CLiMB as a leaderboard, we design it as a framework to study exactly these questions about how such distribution shifts can be dealt with in a CL setting. In fact, our findings with VCR show that task-specific tricks (such as drawing bounding boxes onto images) that can facilitate effective learning on individual tasks do not generalize well to a continual learning setting. Models such as UNITER, which do not use similar tricks because they utilize visual object features rather than image patches, might not show the same VCR-related issue that ViLT does.
>
> [1] *Learn Continually, Generalize Rapidly: Lifelong Knowledge Accumulation for Few-shot Learning* Xisen Jin, Bill Yuchen Lin, Mohammad Rostami, Xiang Ren, Findings of EMNLP 2021
>
> [2] *12-in-1: Multi-Task Vision and Language Representation Learning*, Jiasen Lu, Vedanuj Goswami, Marcus Rohrbach, Devi Parikh, Stefan Lee, CVPR 2020

---

### Official Review · Reviewer_VSFG · 2022-07-28
**Well-motivated benchmark for multimodal continual learning.**

**Rating:** 6
**Confidence:** 2
**Correctness:** Yes.

**Strengths:**

1. To the best of my knowledge, this paper is the first CL benchmark on multimodal tasks. It can facilitate research on multimodal task adaptation and mitigating "catastrophic forgetting".
2. The paper is well-written, and tasks and evaluation metrics are clarified.
3 The experiments provide insightful results, that is existing CL strategies do not generalize well to sequences of multimodal tasks. Such results also demonstrate the necessacity of this benchmark.

**Weaknesses:**

As a benchmark, more experiments (not only ViLT-based model) are expected to be conducted.
The motivation for mitigating "catastrophic forgetting" in the multimodal scenario (between different multimodal tasks) is weak and need further discussion.

**Additional Feedback:**

None.

**Clarity:**

This paper is well written but the copyright of images is expected to be clarified.

**Documentation:**

Yes.

**Ethics:**

No ethical discussion is provided by the authors. It would be desirable for authors to address it.

**Relation To Prior Work:**

To the best of my knowledge, this work properly discusses previous related contributions.

**Summary And Contributions:**

This paper proposes CLiMB, a continual learning benchmark on vision-and-language tasks. CLiMB includes several CL algorithms and a modified ViLT model. It is presented to benefit for CL in multimodal tasks with deployment to multi- and unimodal tasks. The paper is well-motivated and the evaluation metrics are well-clarified.

---

> ### Author Response · Authors · 2022-08-15
> **Response to Reviewer VSFG**
>
> We thank the reviewer for their comments.
>
> **More experiments, beyond ViLT:** We refer the reviewer to General Response #1 - we have now included UNITER into the CLiMB benchmark, and our revision contains preliminary CL experiments with the UNITER model. The final version of our paper will contain the full complement of UNITER experiments [currently running], including a comparison of how the CL trends in our ViLT experiments compare against those seen in UNITER.
>
> **Motivating Multimodal CL:** The reviewer correctly points out that a stronger real-world motivation for the challenge of multimodal continual learning is required. We refer the reviewer to General Response #2.
>
> **Ethical Discussion:** Thanks for raising this concern; our revision contains this ethical discussion in a separate Limitations section (Section 7). We had initially included a brief ethical discussion in the Discussion and Limitations section of our paper, but after reading through the reviews we agree that this should be promoted to a lengthier discussion.

---

### Author Response · Authors · 2022-08-15
**General Response**

We would like to thank all the reviewers for their feedback and suggestions, as well as for recognizing the originality and importance of the multimodal continual learning problem that we introduce. We would like to use the General Response to address some of the reviewers’ common concerns:

**#1: Experiments with More VL Models:**

As several reviewers (**VSFG, iGVT, hZLm**) pointed out, while our paper currently only includes results using ViLT as the base encoder for continual learning, it is desirable that the benchmark include results from more models. We have thus extended our benchmark to include upstream continual learning experiments using the UNITER [1] pre-trained VL model as the base encoder. We have updated our supplementary material (Section E) to include a preliminary set of CL experiments using 3 tasks (VQA->NLVR->SNLI-VE). We are continuing to run full CL experiments for UNITER, and will include a results table analogous to Table 3 in the final version of the paper. We will also include the code for the UNITER model in our benchmark’s GitHub repository for camera-ready. Further, we have included documentation on how researchers can add new models to the benchmark.

**#2: Why Multimodal Continual Learning? (VSFG, fn9P)**

A long-standing goal of AI is the deployment of agents into the real world that can perceive and interact. The current dominant paradigm in vision-and-language is training of models on individual tasks; however, it is expensive to deploy a new copy of the model for every task that we want our models to perform. Multi-task learning [2, 3] is a solution to this, but requires a fixed set of tasks at deployment. However, it is desirable that these agents can continue to learn new functions and solve new tasks in the world even after deployment, while not forgetting to solve tasks they have already learned, which has motivated the field of continual/lifelong learning. For instance, a multimodal dialog agent such as BlenderBot [https://blenderbot.ai/] needs to continually add capabilities related to information retrieval, question answering (both visual and non-visual), and dialog. However, while most CL research has been concentrated around unimodal tasks, not much work has looked at how this continual learning ability is extended when the model has to deal with multiple input modalities, as these agents will have to. The CLiMB benchmark seeks to fill this gap by providing researchers with a benchmark where this ability of multimodal models to learn continually can be systematically studied.

[1] *UNITER: UNiversal Image-TExt Representation Learning* Yen-Chun Chen, Linjie Li, Licheng Yu, Ahmed El Kholy, Faisal Ahmed, Zhe Gan, Yu Cheng, Jingjing Liu, ECCV 2020

[2] *12-in-1: Multi-Task Vision and Language Representation Learning*, Jiasen Lu, Vedanuj Goswami, Marcus Rohrbach, Devi Parikh, Stefan Lee, CVPR 2020

[3] *VL-Adapter: Parameter-Efficient Transfer Learning for Vision-and-Language Tasks*, Yi-Lin Sung, Jaemin Cho, Mohit Bansal, CVPR 2022

---

### Author Response · Authors · 2022-08-25
**Discussion Period ends Aug 29**

Hi all, we just wanted to thank the reviewers again for their insightful feedback and remind everyone that the discussion period ends August 29 (in a few days!). We would be grateful if the reviewers who raised concerns or identified weaknesses in our work can read through our response and our updated draft, as we believe we have addressed many of your comments and strengthened the submission considerably.

---

### Meta-Review · Area_Chair_eNMP · 2022-09-07

**Recommendation:** Accept
**Confidence:** 4

**Metareview:**

Overall, the reviewers have expressed positive support for the submission. It proposes a continuous learning framework for multi-modal models. The reviewers' concerns were quickly addressed in the responses. The primary concern appears to be a weak motivation for the problem being tackled. The authors propose a stronger framing I am hoping to see in the final version of this paper. I am in support of accepting the paper.

---

### Decision · Program_Chairs · 2022-09-16

Accept